# Comparison of Microalgae *Nannochloropsis oceanica* and *Chlorococcum amblystomatis* Lipid Extracts Effects on UVA-Induced Changes in Human Skin Fibroblasts Proteome

**DOI:** 10.3390/md22110509

**Published:** 2024-11-10

**Authors:** Sinemyiz Atalay Ekiner, Agnieszka Gęgotek, Pedro Domingues, Maria Rosário Domingues, Elżbieta Skrzydlewska

**Affiliations:** 1Department of Analytical Chemistry, Medical University of Bialystok, Mickiewicza 2D, 15-222 Bialystok, Poland; sinemyiz.atalay-ekiner@umb.edu.pl (S.A.E.); agnieszka.gegotek@umb.edu.pl (A.G.); 2Mass Spectrometry Centre, LAQV-REQUIMTE, Department of Chemistry, University of Aveiro, Santiago University Campus, 3810-193 Aveiro, Portugal; p.domingues@ua.pt (P.D.); mrd@ua.pt (M.R.D.)

**Keywords:** fibroblast, UVA radiation, *Nannochloropsis oceanica*, *Chlorococcum amblystomatis*, lipid extracts, oxidative stress, inflammation, proteomics, cytoprotective effect

## Abstract

Lipid extracts from the microalgae *Nannochloropsis oceanica* and *Chlorococcum amblystomatis* have great potential to prevent ultraviolet A (UVA)-induced metabolic disorders. Therefore, the aim of this study has been to analyze their cytoprotective effect, focused on maintaining intracellular redox balance and inflammation in UVA-irradiated skin fibroblasts, at the proteome level. The above lipid extracts reversed the suppression of the antioxidant response caused by UVA radiation, which was more visible in the case of *C. amblystomatis*. Modulations of interactions between heme oxygenase-1 and matrix metalloproteinase 1/Parkinson’s disease protein 7/transcript1-α/β, as well as thioredoxin and migration inhibitory factor/Parkinson’s disease protein 7/calnexin/ATPase p97, created key molecular signaling underlying their cytoprotective actions. Moreover, they reduced pro-inflammatory processes in the control group but they also showed the potential to regulate the cellular inflammatory response by changing inflammasome signaling associated with the changes in the caspase-1 interaction area, including heat shock proteins HSP90, HSPA8, and vimentin. Therefore, lipid extracts from *N. oceanica* and *C. amblystomatis* protect skin fibroblast metabolism from UVA-induced damage by restoring the redox balance and regulating inflammatory signaling pathways. Thus, those extracts have proven to have great potential to be used in cosmetic or cosmeceutical products to protect the skin against the effects of solar radiation. However, the possibility of their use requires the evaluation of their effects at the skin level in in vivo and clinical studies.

## 1. Introduction

Ultraviolet (UV) radiation, a part of sunlight reaching the earth, is the main physical factor that poses a daily threat to human skin [1,2]. Due to the wavelength characteristics, UV radiation is classified as follows: ultraviolet A (UVA), B (UVB), and C (UVC) with the UV wavelengths of 315–400, 280–315, and 200–280 nm, respectively [3]. However, UVC is mostly absorbed by the ozone in the atmosphere [3]. Both UVA and UVB radiation reaching the skin may cause metabolic alterations in skin cells that account for the causes of the development of skin diseases including psoriasis, atopic dermatitis [4], and even skin cancers, including melanoma [5]. While UVB penetrates mainly the epidermis and papillary dermis, UVA affects the dermis, namely its full thickness including the subcutaneous fat [6]. Consequently, photoaging in the dermis is marked by a loss of fibroblast density and extracellular matrix alteration, including the accumulation of elastin-rich elastic fibers in the reticular dermis [6]. This effect is mainly mediated by the oxidative stress induced by UV radiation and the associated chronic inflammation, causing the disruption of cellular metabolism [7]. This is particularly evident with UVA radiation that penetrates the dermis layer of the skin, inter alia, triggering oxidative inflammatory reactions. The fibroblasts, major cells of the dermis [1,8], are significantly affected by the UVA damage. Since they play major roles in many biological processes, including intercellular signaling through interactions with other skin cells, such as keratinocytes and melanocytes [9], and the regulation of cell differentiation [10], UVA-induced alterations in fibroblasts constitute a major cause of cellular and skin damage.

Therefore, new pharmacotherapeutic approaches have been increasingly sought to counteract the above-mentioned metabolic alterations of the skin caused by UVA radiation exposure. In order to minimize side effects, significant attention is being focused on substances containing bioactive compounds that modulate intracellular redox signaling and inflammation and that show the potential for large-scale production. In recent years, marine-derived active substances have shown promising potential uses in the treatment of UV damage, such as antioxidant peptide ETT from *Isochrysis zhanjiangensis* [11], peptides from Skipjack tuna cardiac arterial bulbs [12], antioxidant peptides from Skipjack tuna (*Katsuwonus pelamis*) skin [13], collagen peptides from bigeye tuna (*Thunnus obesus*) skin and bone [14], gelatin from cartilage of Siberian sturgeon (*Acipenser baerii*) [15], etc. Lipid extracts obtained from microalgae of various origins have become increasingly popular, including marine species, such as *Nannochloropsis oceanica,* and freshwater species, such as *Chlorococcum amblystomatis*. These extracts contain bioactive lipids (polyunsaturated fatty acids (PUFAs) from phospholipids being the main vector of omega-3 (ω-3) fatty acids and glycolipids) and other lipid-soluble compounds that modulate the cellular redox balance and inflammation [16,17]. The lipid extract from *C. amblystomatis* consists of not only a high content of fatty acids but also carotenoids highlighting the strong antioxidant potential of these extracts [18]. The presence of chlorophyll in the *C. amblystomatis* extract [18] suggests a therapeutic benefit by enhancing tolerance to oxidative stress [19].

Previous work has shown that applying the lipid extract from *N. oceanica* to keratinocytes after UVB irradiation leads to a significant reduction in the levels of sphingomyelin and lysophosphatidylcholine, suggesting an anti-inflammatory and pro-survival effect [16]. Additionally, another study has demonstrated the protective effects of the *N. oceanica* lipid extract in redox-dependent skin cell metabolism, including anti-inflammatory effects due to the reduction in the level of tumor necrosis factor alpha (TNFα), 8-iso prostaglandin F2, and 4-hydroxynonenal (4-HNE)-protein adducts [20]. A study comparing thirty-two different marine microalgae lipid extracts also revealed the powerful antioxidant and anti-inflammatory action of the *N. oceanica* ethanol lipid extract via decreasing lipopolysaccharide (LPS)-induced inducible nitric oxide synthase (iNOS) and cyclooxygenase-2 (COX-2) expression and reducing the reactive oxygen species (ROS) and malondialdehyde (MDA) levels [21]. Moreover, given the content of polar lipids in the *C. amblystomatis* lipid extract, the ability to increase the scavenging of free radicals (exhibited by 1,1-diphenyl-2-picrylhydrazyl (DPPH) and 2,2′-azino-bis-3-ethylbenzthiazoline-6-sulphonic acid (ABTS) radical scavenging assays) as well as the inhibition of human COX-2 have also been demonstrated [22]. 

A recent study comparing the metabolic effects of *N. oceanica* and *C. amblystomatis* on UVA-treated skin fibroblasts highlighted the enhancement of the cellular antioxidant response by reducing the UV-induced ROS levels and increasing the nuclear factor erythroid 2-related factor 2 (Nrf2) expression as well as the activity/level of cellular antioxidant proteins [23]. It was manifested by a reduction in the level of protein-4-HNE adducts and protein carbonyl groups. The action of the *N. oceanica* extract promoted a significant reduction of enzymatic lipid metabolism assessed by the level of eicosanoids and endocannabinoids [23].

Despite the promising related literature data indicating the cytoprotective effect of lipid extracts from *N. oceanica* and *C. amblystomatis*, there is still a lack of analyses of the molecular signaling pathways involved in the effects of microalgae. Therefore, this study aims to evaluate the changes in the proteomic profile of human skin fibroblasts exposed to UVA and treated with the *N. oceanica* or *C. amblystomatis* lipid extracts. Since oxidative stress and inflammation are the two close dynamic processes interplaying one another [24,25], particular emphasis will be put on the interaction of the proteins involved in the antioxidant and cellular inflammatory response against proteome disruption by the UVA-mediated oxidative stress [8]. In order to explore the inflammatory response in greater detail, we will examine the caspase-1 interactome focusing on its interaction with inflammasome complexes, which are the key components of inflammation [26], under the conditions of oxidative stress generated by UVA irradiation. This approach aims to shed light on the molecular mechanism underlying intracellular redox signaling and inflammation, particularly through the protein–protein interactions.

## 2. Results

This study presents the results of the experiment aimed at comparing the protective effects of lipid extracts from microalgae *N. oceanica* and *C. amblystomatis* against the UVA-induced changes in the proteome of human skin fibroblasts. The compared cell groups with their abbreviations used are described in Table 1. In this study, a total of 494 proteins have been identified in the fibroblast proteome. The list of identified proteins (their names, Uniprot IDs, peptide counts as razor + unique, and unique as well as average intensity levels) has been shared in Appendix A. According to the one-way ANOVA, 309 of the proteins (62.55%) have been identified with a significantly altered protein intensity in all the experimental groups (Figure 1, Appendix A). 

We have observed that the proteins with significantly altered the expression levels were primarily categorized into the following groups, listed in descending order based on their prevalence: the proteins involved in the regulation of protein homeostasis (15.86% of the proteins had significantly changed expression levels), the proteins participating in the regulation of the cellular inflammatory response (9.71% of the proteins had significantly changed expression levels), the proteins involved in cellular antioxidant defense (6.15% of the proteins had significantly changed expression levels) and the proteins participating in the regulation of cell survival and differentiation (4.53% of the proteins had significantly changed expression levels). Among all the experimental groups, it has been observed that in the case of topical treatment with the *N. oceanica* or *C. amblystomatis* lipid extracts following UVA irradiation as well as UVA irradiation alone, the most affected protein group—in terms of the protein expression level—was the group of protein homeostasis. These proteins are involved in the regulation of the balance of cellular production, folding, and the degradation of proteins [27] in the fibroblast proteome.

Given the small sample size (n = 3), the results of the PCA should be interpreted with caution, as the observed variability between the samples may limit the robustness of the PCA in accurately capturing the full spectrum of the experimental variation. However, in our study, the 3D principal component analysis (PCA) of the proteomic data has shown good clustering of the CTR and UVA samples in distinct groups (Component 1—36.2%; Component 2—12%; Component 3—6.7%; Figure 2). On top of that, the treatment with the lipid extracts from *N. oceanica* (alone) or *C. amblystomatis* (following UVA irradiation) favored a better clustering pattern as compared to the *C. amblystomatis* lipid extract treatment or the treatment with the lipid extract from *N. oceanica* after UVA irradiation. The treated groups (*N.o.*, *C.a.*, UVA + *N.o.*, and UVA + *C.a.*) have displayed distinct protein expression profiles as compared to the UVA-only group. Furthermore, volcano plots have shown significant differences in the proteome of fibroblasts when comparing the effects of the *N. oceanica* or *C. amblystomatis* lipid extracts on the control fibroblasts as well as UVA-irradiated fibroblasts (Figure 3). 

A heatmap was generated using the top 25 proteins with a significantly changed expression in the skin fibroblasts. The upper hierarchical dendrogram showed that the samples were clustered into distinct groups: the control group (**CTR**), the group corresponding to the cell treated with the *N. oceanica* or *C. amblystomatis* lipid extracts (***N.o.***/***C.a.***), the group of UVA-irradiated cells (**UVA**), and the group of the cells treated with the *N. oceanica* or *C. amblystomatis* lipid extracts following UVA irradiation (**UVA + *N.o.***/**UVA + *C.a.***) (Figure 4). In parallel, the dendrogram created using all the identified proteins showed an apparent hierarchical clustering of the samples from the fibroblast cell groups (Appendix A). The CTR group was clustered independently from all the other cell groups. In addition, treatment with the *N. oceanica* or *C. amblystomatis* lipid extracts significantly altered the protein expression pattern caused by UVA irradiation. Interestingly, although the **UVA + *N.o.*** group clustered closely to the **UVA** group, the application of the *C. amblystomatis* lipid extract after UVA irradiation (**UVA + *C.a.***) resulted in a protein profile that was more distantly clustered from the UVA group as compared to the **UVA + *N.o.*** group.

Another heatmap was created with a special emphasis on the proteins with a significantly changed expression involving the regulation of the cellular inflammatory response, protein homeostasis, cellular antioxidant defense, or cell survival and differentiation. This heatmap has shown the *N. oceanica* or *C. amblystomatis* lipid extracts to effectively prevent the changes in the protein expression caused by UVA irradiation (Appendix A). Based on the changing protein profiles of fibroblast groups, as shown in the heatmaps, treatment with the lipid extracts from *N. oceanica* or *C. amblystomatis* primarily impacted the proteins involved in the regulation of structural integrity, catalysis, particularly those influencing the redox balance and inflammatory responses, and molecular chaperones responsible for maintaining protein homeostasis (Figure 4, Appendix A).

The fold changes, calculated using the average protein intensities (Table 2), have shown the treatment with the *N. oceanica* or *C. amblystomatis* lipid extracts to have modified the expression of antioxidant and cytoprotective proteins, in response to the UVA irradiation-induced oxidative stress [8]. In general, it has been observed that UVA irradiation suppressed the cellular antioxidant response, by downregulating the expression level of antioxidant proteins. This trend has been exemplified by the decrease in heme oxygenase-1 level (P09601) after UVA irradiation (Table 2). Treatment with the *C. amblystomatis* lipid extract increased the UVA-dropped heme oxygenase-1 level more than 200-fold. However, there was also alteration in this protein intensity in the case of the treatment with the *N. oceanica* lipid extract after UVA irradiation. In addition, without the UVA exposure, the level of heme oxygenase-1 in was lower in the *C. amblystomatis* lipid extract treated with fibroblasts versus control cells. This should be taken into consideration for future studies including in the multidimensional monitoring of cellular redox as well as cell survival status. Since the cellular antioxidant response is a complex and very dynamic biological process involving numerous proteins [28], it would not be correct to comment that the application of *C. amblystomatis* alone negatively affects the antioxidant response of control cells based on only dramatic changes in the heme-oxygenase 1 expression. Moreover, we have observed the level of peroxiredoxin-5 to have increased due to the application of *C. amblystomatis* alone.

On the other hand, it has been seen that a 9.71-fold increased level of aldo-keto reductase family 1 member A1 (P14550), due to the UVA effect, decreased significantly with both lipid extract treatments (following UVA irradiation). The *C. amblystomatis* lipid extract reduced the UVA-increased level of aldo-keto reductase family 1 member A1 by 95%. In addition, the UVA-increased levels of quinone oxidoreductase (Q08257), peroxiredoxin-5, mitochondrial (P30044), NADH-cytochrome b5 reductase 3 (P00387), as well as aldo-keto reductase family 1 member C1 and B1 (Q04828 and P15121, respectively), were attenuated by the *C. amblystomatis* or *N. oceanica* lipid extract treatment. It is plausible to state that both of the lipid extracts proved a protective effect on the regulation of the redox balance disrupted by UV irradiation. The proteomic dataset revealed the largest changes in proteins with the cytoprotective activity against oxidative stress, as illustrated in Figure 5.

Inflammation and intracellular redox signaling constitute two interrelated dynamic processes that influence one another and play a critical role in critical biological functions, such as pathophysiological changes including alterations of cell survival and differentiation dynamics [24,25]. Therefore, in parallel with the changes in the antioxidant protein profile induced by the *N. oceanica* and *C. amblystomatis* lipid extracts mentioned above, the antioxidant and anti-inflammatory potentials of which have been noted in the related literature [18,21], attention was also given to the expression levels of proteins involved in regulating the cellular inflammatory response.

The level of the CD44 antigen (P16070) in skin fibroblasts has been found to increase approximately 6.93-fold due to the effect of UVA (Table 2). Treatment of fibroblasts with the *N. oceanica* or *C. amblystomatis* lipid extracts, even without the UVA exposure, has produced similar results. However, treatment with the lipid extracts following UVA irradiation, has demonstrated the potential to limit the elevated expression of CD44 induced by UVA. In addition, the UVA-enhanced level of ATP-citrate synthase (P53396), which is involved in the “citrate pathway” participating in the regulation of inflammation by modulating the synthesis of the inflammatory mediators [29], was slightly reduced through the treatment with the *N. oceanica* or *C. amblystomatis* lipid extracts, similarly to what has been observed for CD44. The expression of interstitial collagenase/matrix metalloproteinase-1 (P03956) was also markedly reduced in the UVA-irradiated fibroblasts versus the CTR group (with an almost 5-fold level of change), while the UVA-decreased level was notably increased through the treatment with the *N. oceanica* or *C. amblystomatis* lipid extracts (Table 2). 

Furthermore, UVA notably reduced the level of the macrophage migration inhibitory factor (P14174) in skin fibroblasts. However, this protein was not identified in the fibroblast groups UVA + *N.o.* or UVA + *C.a*. Interestingly, the UVA exposure also greatly increased the levels of dipeptidyl peptidase 4 (P27487) and signal transducer and activator of transcription 1-alpha/beta (P42224), with changes ranging from approximately 16- to 102-fold. Treatment with the *N. oceanica* or *C. amblystomatis* lipid extracts slightly increased or decreased those levels, respectively. Overall, it has been found that treatment of fibroblasts with the *N. oceanica* or *C. amblystomatis* lipid extracts also elevated the expression of those proteins under consideration. 

The UVA-increased level of transitional endoplasmic reticulum ATPase (P55072) has also been found to slightly decrease due to the effect of the *N. oceanica* or *C. amblystomatis* lipid extract. The calnexin (P27824) has been found to increase almost 5-fold in the UVA-irradiated fibroblasts as compared to CTR cells. The treatment with the *N. oceanica* or *C. amblystomatis* lipid extracts slightly increased this expression (1.2-fold). In general, similar to the profile of cytoprotective proteins summarized above, the treatment with the algae lipid extracts has appeared to support the restoration of the UVA-altered expression of proteins involved in the regulation of the inflammatory response (Table 2, Figure 6). However, it is clear that the application of this treatment alone without UVA irradiation also affects the profiles of these proteins.

The heatmap displays the Pearson correlation matrix of the identified proteins with the significantly altered expression between cell groups, specifically those involved in regulating the intracellular redox balance or inflammatory cellular responses (Table 2, Appendix A). The heatmap demonstrates that the proteins mentioned may correlate in expression due to indirect relationships, rather than direct interactions, due to shared regulatory factors such as transcription factors or signaling pathways. The following proteins had the strongest correlation coefficients (correlation cut of >0.97): Calnexin (P27824)–CD44 antigen (P16070), r = 0.971; CD44 antigen (P16070)–signal transducer and activator of transcription 1-alpha/beta (P42224), r = 0.98169; CD44 antigen (P16070)–transitional endoplasmic reticulum ATPase (P55072), r = 0.97593; signal transducer and activator of transcription 1-alpha/beta (P42224)–transitional endoplasmic reticulum ATPase (P55072), r = 0.97372; aldo-keto reductase family 1 member B1 (P15121)–NADH-cytochrome b5 reductase 3 (P00387), r = 0.98562; glutathione S-transferase P (P09211)–galectin-1 (P09382), r = 0.97623; ATP-citrate synthase (P53396)–hexokinase-1 (P19367), r = 0.97282; and peroxiredoxin-4 (Q13162)–glutathione S-transferase omega-1 (P78417), r = 0.97046. On top of this, the proteins with the strongest negative correlation coefficients are as follows: signal transducer and activator of transcription 1-alpha/beta (P42224)–glutathione S-transferase P (P09211), r = −0.87161; macrophage migration inhibitory factor (P14174)–dipeptidyl peptidase 4 (P27487), r = −0.86601; and dipeptidyl peptidase 4 (P27487)–Ras-related protein Rap-1A (P62834), r = −0.86489.

The protein–protein interaction network has revealed cytoprotective proteins against redox imbalance and proteins involved in regulating the inflammatory response, which exhibited the highest fold changes between the analyzed fibroblast groups, demonstrating a well-defined interaction network (at least medium confidence level, as analyzed using the stringApp in Cytoscape) (Appendix A). Based on this network, thioredoxin (interacting with macrophage migration inhibitory factor, Parkinson’s disease protein 7, calnexin, and transitional endoplasmic reticulum ATPase) and heme oxygenase-1 (interacting with matrix metalloproteinase-1, Parkinson’s disease protein 7, signal transducer, and activator of transcription1-alpha/beta) have been found to be two antioxidant proteins linked to more than two proteins involving the regulation of inflammation. Parkinson’s disease protein 7 has appeared to be the protein that interferes with the largest number of antioxidant proteins in this network.

It is also important to highlight the significantly changed proteins that are involved in the regulation of cell survival and differentiation or protein homeostasis. The expression of fibroblast activation protein-α (FAP), a member of the prolyl peptidase family (Q12884), was significantly increased by UVA irradiation as compared to the control cell group. Interestingly, the *N. oceanica* lipid extract treatment following UVA irradiation increased that level almost 21 times whereas the effect of the *C. amblystomatis* lipid extract treatment after UVA irradiation was much less elevated (almost 1.4-fold) (Table 2). 

Additionally, UVA radiation had a pronounced effect on the protein profiles especially in relation to the regulation of protein homeostasis. As shown in Table 2, one of the most significant fold change levels has been observed in the expression profile of heat shock protein HSP 90-beta (P08238). Its level increased by 41.47-fold due to the UVA exposure as compared to the CTR group. The treatment with the *N. oceanica* or *C. amblystomatis* lipid extracts significantly decreased this level. A similar observation has been made in the case of heat shock protein HSP 90-alpha (P07900), with a relatively lower fold change level. In addition, the treatment with the *N. oceanica* or *C. amblystomatis* lipid extracts has also been found to slightly decrease or increase the UVA-induced level of other chaperon proteins and endoplasmic reticulum chaperone BiP (P11021), respectively. 

These lipid extracts significantly induce the upregulation of small ribosomal subunit proteins uS15 and uS4 (P62277 and P46781, respectively), the levels of which were greatly reduced by UVA irradiation. Furthermore, the UVA-increased levels of 26S proteasome non-ATPase regulatory subunit 2 (Q13200) and Cullin-associated NEDD8-dissociated protein 1 (Q86VP6) dropped through the treatment with the *N. oceanica* or *C. amblystomatis* lipid extracts. The application of these two lipid extracts alone significantly altered the profiles of proteins involved in both cell survival and differentiation as well as the regulation of protein homeostasis in skin fibroblasts.

In order to understand the molecular action behind the lipid extracts’ protective activity in the protein interplay regarding inflammasome complexes, the key regulators of the cellular inflammatory response [30] associated with the inflammasome activity [31], the caspase-1 interactome area was additionally examined by means of the IP-supported MS analysis. This analysis was performed in addition to the main proteomic analysis, focusing on the primary proteomic data seeking the relationship between the inflammation and redox imbalance at the protein level presented above, which is the main focus in this article. The limited data, with only 10 identified proteins, hinder a reliable statistical analysis, making it difficult to draw definitive conclusions and providing only qualitative information, rather than a quantitative one. Nevertheless, in order to avoid any data missing, the list of the proteins has been shared in Appendix A (their name, ID, number of associated peptides, number of unique peptides, and average intensity level). Importantly, heat shock protein HSP 90-beta (P08238) has been found to be in the interactome area of caspase-1. This has drawn attention to the dramatic change in the same protein level observed in the whole proteomic data (Table 2), as we have mentioned above. This confirms the importance of this protein in the modulation of the molecular activity induced by the lipid extracts in the case of UVA-irradiated fibroblasts. Other than that, the following interesting proteins have also been identified, with different intensities within cell groups: vimentin (P08670), heat shock cognate 71 kDa protein (P11142), pyruvate kinase PKM (P14618), interferon-induced, double-stranded RNA-activated protein kinase (P19525), and E3 ubiquitin-protein ligase RNF213 (Q63HN8).

## 3. Discussion

Daily skin exposure to UV radiation, mainly UVA (315–400 nm) and UVB (280–315 nm), is one of the main harmful factors causing skin diseases such as photoaging, psoriasis, atopic dermatitis, and even skin cancer due to interfering with critical metabolic processes such as inflammation, cell aging and death, DNA damage/repair, and redox homeostasis [4,5,7]. 

UVB has been shown to increase the level of membrane-bound nitric oxide synthase (NOS) and associated nitric oxide (NO) generation in human keratinocytes, while UVA, reaching fibroblasts, may induce the generation of superoxides, to inactive iron–sulfur proteins leading to the release of the reducing ferrous iron, and ultimately, resulting in further production of ROS [7]. UVA irradiation may also promote the activation of nuclear factor kappa, NF-κB, (via activation of mitogen-activated protein kinases, MAPKs, such as extracellular signal-regulated kinase, ERK, and c-jun N-terminal kinase, JNK, associated with the downstream activation of transcription factor activator protein 1, AP-1) and thus, may modulate critical biological processes such as inflammation, cell cycle progression, proliferation, and apoptosis [7]. With the increase in UV exposure due to changing climate conditions, there is an increasing need to explore new, effective therapeutic strategies with minimal or tolerable side effects to prevent oxidative damage in the skin caused by UVA irradiation [32]. At the proteome level, our study confirms the beneficial activity of the topical treatment with *C. amblystomatis* or *N. oceanica* lipid extracts [16,17], against the perturbed protein expressions in skin fibroblasts due to UVA irradiation (Figure 7). This supports the added value of these lipid extracts and their associated therapeutic potential. 

Lipid exposure rapidly alters protein expression through the complex signaling cascades (activation of kinase cascades; receptor-mediated signaling such as G-protein-coupled receptor activity changes; membrane and stress responses (alteration of membrane composition and associated changes in intracellular signaling such as transcriptional regulation); unfolded protein response according to endoplasmic reticulum stress; post-transcriptional modifications (such as lipidation, phosphorylation, and ubiquitination of proteins), and targeted protein degradation) [33,34,35,36,37]. Together, these mechanisms enable cells to respond dynamically to lipid levels, shifting protein expression profiles to maintain intracellular homeostasis or adapt to stress conditions. Our study highlights that the interplay between the regulation of intracellular redox status and inflammatory cell response appears to be the key molecular regulatory action responsible for the protective effect of these lipid extracts on the UVA damage mediated by the altered metabolism of skin fibroblasts (Figure 7). 

### 3.1. Restoring Intracellular Redox Balance Disrupted by UVA Irradiation

The changes in intracellular redox signaling due to oxidative stress and inflammatory signaling are two close biological processes interacting with one another [24,25]. These dynamic changes concurrently affect cell survival and even the success of antioxidant therapies, the limitations and failures of which have been indicated in various in vivo and clinical studies [38]. Therefore, a key focus of this study has been to investigate the ability of these two lipid extracts to modulate the UVA-induced disruption of the redox balance and inflammatory response at the proteome level. Unraveling the complex molecular mechanisms underlying the cytoprotective effects of these lipid extracts paves the way for developing effective, targeted therapeutic approaches for diseases associated with oxidative stress.

This study reveals that the use of the *N. oceanica* or *C. amblystomatis* lipid extracts promotes the rebalancing of the protein expression and related pathways disturbed by UVA radiation by regulating intracellular redox balance and inflammation. The results gathered in our study demonstrate that these two lipid extracts are capable of restoring the cytoprotective protein expression disrupted by the UVA exposure. This effect is especially evident in the case of heme oxygenase-1 (HO-1) and aldo-keto reductase family member 1 A1 (AKR1A1), proteins with critical cytoprotective effects against oxidative stress, especially after using the *C. amblystomatis* lipid extract [39,40]. This is in accordance with the related literature, previously referred to, that has described the potential protective roles of these extracts in regulating the redox balance and intracellular inflammatory status due to their high content of bioactive lipids [16,17,22].

In addition, our study indicates that the expression of other antioxidant/cytoprotective proteins is also regulated by the *N. oceanica* or *C. amblystomatis* lipid extracts applied after the exposure to UVA radiation. However, this effect was observed as a relatively milder change compared to the case of HO-1 and AKR1A1. A recent study of skin keratinocytes irradiated with UVA has shown that both lipid extracts (at the same concentration) significantly reduce the level of ROS increased by UVA and increase the expression of Nrf2 along with the increased activity/level of antioxidants (superoxide dismutase 1/2, SOD1/2, catalase, CAT, glutathione, GSH, thioredoxin, and Trx), and concurrently, with the decrease in the level of the highly reactive lipid peroxidation product, 4-HNE, and its protein adducts [23]. Taking into account the methodological differences in the measurement of protein levels (ELISA method [23] versus mass spectrometry used herein and also characterized by higher sensitivity), it should be noted that the observed mild effect on the overall antioxidant protein expression may be a result of the concentration used and/or time-dependent changes in cellular response and needs further evaluation. The concentration used and the 24 h application may have initially lowered ROS levels and reduced lipid peroxidation, as observed in [23]. However, this duration may not have fully impacted the protein expression and the maturation of the relevant proteins.

When the effectiveness of these two lipid extracts on the antioxidant response of fibroblasts was compared, the *C. amblystomatis* lipid extract appeared to be more effective in reversing the UVA-suppressed antioxidant response at the proteome level, although the overall effect was close. Given the dramatic increase in the HO-1 level that is reduced by UVA irradiation, the antioxidant/cytoprotective action of the *C. amblystomatis* lipid extract has been found to be centered on the HO-1 pathway, which proves an important role in skin injury and wound healing due to its antioxidative and anti-inflammatory effects [41]. HO-1 mediates the suppression of signal transducer and activator of transcription 3 (STAT3), which is critically involved in the pathogenesis and development of psoriasis by promoting inflammation and cell differentiation. This suppression occurs via Src homology phosphatase-1 (SHP-1) signaling, suggesting a potential therapeutic application for HO-1 in psoriasis [41]. The cytoprotective activity of HO-1 and heme degradation-end products, biliverdin, carbon monoxide (CO), and ferrous (Fe^2+^) ions, present great potential for the development of clinical applications against inflammatory and oxidative injury [42].

### 3.2. Regulating Inflammation Through Alterations in the Intracellular Redox Balance

Interestingly, the results gathered in our study have revealed the interaction between HO-1 and matrix metalloproteinase-1 (MMP1), Parkinson’s disease protein 7 (PARK7), and signal transducer and activator of transcription1-alpha/beta (STAT1) that display important roles in inflammation [43,44,45]. The related literature generally indicates that UV irradiation induces the expression of matrix metalloproteinases (MMP1, MMP3, and MMP9) in skin fibroblasts [46]. However, our study has found that the level of MMP1 is significantly reduced in response to UVA exposure. This may be a dose-dependent result of UVA exposure leading a protective or adaptive response in fibroblasts by reducing MMP1 expression (protective cell adaptation at lower levels of stress—potentially—which may limit the cellular damage by reducing extracellular matrix breakdown) [47]. It may have also occurred because of the exposure duration of UVA. Because MMP1 exposure might be differentiated by a specific time-course response (an initial reduction followed by an increase in expression later, due to the dynamic interaction with tissue inhibitor of metalloproteinase, TIMP, [48]). In addition, it should be remembered that UVA can induce oxidative stress but also activate cellular antioxidant response [49]. Therefore, if the UVA exposure applied in our study triggered a relatively mild/low-level stress, this should also be taken into account. Further experiments including analyzing ROS levels or evaluating antioxidant and transcription factor activity could help to clarify this situation. The downregulation of MMP2 and MMP9 has also been demonstrated in epidermal keratinocytes 24 h after UVA irradiation [50]. However, it is known that an increase in the HO-1 expression is generally accompanied by a decrease in the MMP1 expression [51]. The differences observed in the data presented in this study compared to the related literature (as seen in the MMP-1 level) may be a result of not only the duration after UVA irradiation (difference in the acute and late cellular response) as mentioned above but also the oxidative stress severity.

Complex redox signaling may also alter the MMP1 expression. A recent study indicates that activation of the extracellular signal-regulated kinase (ERK) signaling may also result in the MMP1 transcriptional activation in the peroxiredoxin 3 (PRDX3)-overexpressed breast cancer cells [52]. Thus, the antioxidant response suppressed by the UVA may also have affected the MMP1 expression. Unlike in the case of PRDX5, the results obtained indicate that UVA reduces the level of PRDX1/2/4/6. The treatment of fibroblasts with the two types of algal lipid extracts induces similar reductions in the MMP1 levels under the influence of UVA. It may indicate that both extracts may support the healing process accompanied by inflammatory signaling in the UVA-irradiated skin fibroblasts. In the wounded skin, MMP1, known as collagenase, is expressed at the wound site during that particular phase of wound healing and is involved in chemokine signaling [53]. The results obtained indicate that the application of the algae lipid extracts to the control group of fibroblasts reduces the level of MMP1, which suggests that the use of those extracts may also reduce pro-inflammatory processes in skin fibroblasts not under oxidative stress conditions.

Our study shows that not only HO-1 but also PRDX5, TRX, and quinone oxidoreductase interact with PARK7 to participate in the cellular response to UVA irradiation as well as the treatment with the *N. oceanica*/*C. amblystomatis* lipid extracts. PARK7 protein is a non-standard protein showing high antioxidant activity and the ability to stabilize the Nrf2 transcription factor, and consequently, inducing SOD1 [44], which is considered to be associated with autosomal recessive Parkinson’s disease. In addition to its antioxidant effect, thanks to its protective activity, PARK7 may also reduce abnormal protein aggregation and inhibit the production of TNF-α and interleukin 4, IL-4, in mast cells [44]. Our findings indicate that PARK7 may be involved in the metabolic effects of the *N. oceanica*/*C. amblystomatis* lipid extracts by regulating the redox balance disturbed by UVA at the level of the antioxidant protein expression. The related literature suggests, among others, a positive correlation between the PARK7 activity and the HO-1 expression [54]. Therefore, the UVA-induced reduction in the HO-1 and PARK7 levels in this study confirms this relationship. The PARK7 expression has been found to be slightly reduced in the cells treated with the *N. oceanica*/*C. amblystomatis* lipid extracts after UVA irradiation, as compared to the UVA-exposed cells. It may be a result of complex redox signaling triggered by the redox modulatory effect of both lipid extracts, including the regulation of the other antioxidant proteins’ expression and activity as well as the activity of several other chaperon proteins [55]. 

Additionally, the results indicate that the interaction between HO-1 and STAT1 plays a crucial role in the molecular mechanisms underlying the cellular response to the algal lipid extracts. The activation of STAT1 by TLR (Toll-like receptor) signaling involves the TLR-induced innate immune response [45]. Furthermore, STAT1 has been suggested to be a key mediator of the proinflammatory response of resident microglia and infiltrating blood-borne macrophages in traumatic brain injury [56]. On the other hand, CO (carbon monoxide), produced by the oxidative catabolism of heme via the HO-1 activity, is involved in CD8+ T-cell effector function through modulating STAT1/3 signaling [57]. Our study has revealed that both lipid extracts are able to reduce the level of STAT1, which increases dramatically due to UVA irradiation. In addition to highlighting the anti-inflammatory effects of both lipid extracts, these findings may also underscore the significance of STAT1 signaling in the molecular activity of the extracts, particularly in relation to HO-1. However, it is important to note here that the algae lipid extracts used, without UVA irradiation, increase the STAT1 level in skin fibroblasts almost as much as UVA does. This reveals the need for further detailed evaluation of the topical application of the aforementioned lipid extracts to healthy skin cells, regarding inflammation and cell survival dynamics [58].

Likewise, given the interplay of inflammation and redox signaling, TRX has been found to interact with the macrophage migration inhibitory factor (MIF), PARK7, calnexin, and transitional endoplasmic reticulum ATPase. MIF, as a pro-inflammatory cytokine, is known to involve pathophysiology related to inflammation and fibrosis [59]. Calnexin, apart from its chaperone function in the ER, has been also suggested to be linked to the central nervous system endothelial cells driving neuroinflammation [60]. The transitional endoplasmic reticulum ATPase p97, known as VCP (valosin-containing protein), facilitates the alternative NF-κB signaling pathway activation due to the p97-nuclear protein localization protein 4 homolog (Npl4)-ubiquitin-dependent degradation pathway (Ufd1) complex regulating the degradation of p100 [61]. In this study, both lipid extracts have displayed the potential to reverse the UVA-mediated alteration of those proteins (it is more obvious in the case of MIF and VCP). However, this effect is not observed as dramatic as their action on the expression of HO-1 and the interaction with MMP1, PARK7, and STAT1. Thus, the alteration of MIF, calnexin, and VCP signaling through the treatment with both lipid extracts (particularly in the case of the *C. amblystomatis* lipid extract) may present a minor effect as a part of the redox-mediated inflammatory response to the UVA irradiation-mediated redox imbalance in skin fibroblasts.

Importantly, our data have shown that UVA irradiation dramatically suppresses the expression of the ribosome subunits identified, including small ribosomal subunit protein uS5, eS19, uS9, uS7, uS15, uS4, eS8, eS8, and uS3 and large ribosomal subunit protein uL1, eL8, and uL10. This reduction in protein translation results from oxidative stress induced by UVA irradiation [62]. Ribosomes are critical targets for oxidative stress, and the functionality of chaperones appears as one of the key cellular responses to maintain translational capacity and, ultimately, protein homeostasis during oxidative stress [62]. In our study, the cellular response to UVA-mediated oxidative stress has been accompanied by a significant increase in the expression of chaperones (endoplasmic reticulum chaperone BiP; heat shock protein HSP 90-alpha and HSP 90-beta; putative heat shock protein HSP 90-beta 2; and heat shock 70 kDa protein 4). The expression and activity of molecular chaperones are increased during oxidative and associated proteotoxic stress due to their role in assisting newly synthesized proteins to reach their native fold and maintaining either their assembly or unfolding and disassembly while protecting them from oxidative stress and protein aggregation [55]. The reduction in the ribosomal protein expressions and the increase in the expression of chaperones, reduced by the UVA irradiation, are remarkably reversed by the treatment with the *N. oceanica* or *C. amblystomatis* lipid extracts. Here, in this regard, the activity of the *C. amblystomatis* lipid extract is found to be more noticeable as compared to the *N. oceanica* lipid extract. On top of this, our data indicate that both lipid extracts may help alleviate UVA-induced oxidative stress, and therefore, maintain protein translation, by protecting the cell against the formation of protein aggregates, in skin fibroblasts.

Although the lipid composition of both lipid extracts shows similarity, which is to say, both are enriched with polyunsaturated omega-6 and omega-3 fatty acids (PUFAs) showing antioxidant activity and anti-inflammatory potential, there are differences between lipidomic profiles [16,22]. It can be speculated that although the differences of the effectiveness observed between the two lipid extracts may be coming primarily from this lipidomic variation, it is recommended that a full quantitative content analysis should be performed in future studies to focus on the cause of this difference.

In order to better understand the inflammatory effects resulting from the changes in protein levels under the UVA exposure to cells, in addition to the main proteomic studies discussed above, the interaction area of caspase 1 with other proteins has also been analyzed qualitatively, rather than quantitatively. It has been found that in the case of the UVA-exposed fibroblast therapy with the use of the lipid extracts from *N. oceanica* or *C. amblystomatis,* there is a change in the protein profile associated with caspase-1 and its interactions with inflammasome complexes (NLRP1, NLRP3, NLRC4, and AIM2) that are the key molecules driving the inflammatory response [31]. It may also be important as an indirect regulatory effect involving the cytoprotective effects of the lipid extracts under consideration.

Importantly, our study has indicated that heat shock proteins HSP 90-beta (HSP90), heat shock cognate 71 kDa protein (HSPA8/HSC70), and vimentin have been found to be associated with the caspase-1 interaction area. HSP90 is suggested to bind NLRP3 and regulate the inflammasome activation and the associated interleukin (IL)-1β release [63]. Moreover, HSPA8/HSC70 may interact with NLRC4, and decreasing this interaction has been found to promote the caspase-1 activation [64]. Furthermore, vimentin, a type III intermediate filament, has been suggested as a key regulator for the NLRP3 inflammasome activity through direct protein–protein interaction by showing the decreased caspase-1 activity in Vim^−/−^ and vimentin-knockdown macrophages [65]. In the first place, our data support the regulatory action of HSP90, HSPA8, and vimentin in the caspase-1-associated inflammasome activity, presenting their identification in the caspase-1 interactome area. These data may also indicate that the application of the *N. oceanica* and *C. amblystomatis* lipid extracts after UVA irradiation of skin fibroblasts may promote the modulation of HSP90, HSPA8, and vimentin signaling, as well as the caspase-1 activity, which consequently modifies the cellular inflammatory response associated with these proteins.

## 4. Materials and Methods

### 4.1. Microalgae Spray-Dried Biomasses 

The marine microalgae *N. oceanica* and freshwater microalgae *C. amblystomatis* were supplied as spray-dried (powder) form by the company Allmicroalgae, Natural products S.A. located in Rua 25 de Abril s/n 2445-413 Pataias, Portugal. They were cultivated in Guillard’s F2 culture medium adapted to the local water [66] supplemented with magnesium and NaCl with a salinity of 30 g/L. And they were grown for 7–15 days in 5 L flask reactors with continuous exposure to light at 700 µmol photons × m^2^·s^−1^. Five flask reactors (5 L) were used to seed one outdoor flat panel (FP) reactor with a capacity of 0.1 m^3^L which will be later expanded to 1 m^3^ FP. As inoculum, 4 FPs were used in a 10 m^3^ tubular photobioreactor (PBR). The PBR was exposed to ambient light and temperature (controlled via a sprinkler-style irrigation system to take the temperature below the maximum limit) with pH stability (maintained via pulsed CO_2_ injections) until the stationary phase was reached. Upon reaching an amount of approximately 50 g·L^−1^, microalgae were dried by atomization in a spray dryer (evaporation efficiency 150 kg of water per 1 h). And, quickly, the obtained microalgae were dried in an air stream (at a temperature of 215 ± 5 °C). The temperature of the outlet air with the biomass powder was 92 ± 3 °C. The obtained powder was stored in a place using a cyclone protected from light and moisture. These spray-dried *N. oceanica* and *C. amblystomatis* microalgae biomass used for lipid extraction. 

### 4.2. Lipid Extraction 

Lipids were extracted from the mentioned *N. oceanica* and *C. amblystomatis* microalgae biomass using a modified Folch method, previously described [67,68]. Briefly, the extraction process was performed using a solvent mixture of dichloromethane: methanol (2:1, *v*/*v*) which was added to 25 mg of biomass. After centrifugation of the samples (670× *g*, 10 min), the supernatant was collected and the residual biomass was subjected to repeated extractions three more times. The combined supernatants were dried under a nitrogen stream; then, they were dissolved in dichloromethane and methanol, and well-vortexed. Following the addition of milliQ water and centrifuging (670× *g* for 10 min), the organic phase was collected (the aqueous phase was re-extracted twice more). The lipid extract was a combination of all organic phases, and the lipid content of the extract was determined gravimetrically. The characterization of the lipid profiles of the *N. oceanica* and *C. amblystomatis* extracts was performed by liquid chromatography coupled with high-resolution mass spectrometry (HILIC-MS) and tandem MS (MS/MS) using a Q Exactive hybrid quadrupole Orbitrap mass spectrometer (Thermo Fisher Scientific, Bremen, Germany), as reported previously [68].

### 4.3. The Method for Cell Culture and the Preparation of Experimental Fibroblast Groups

Skin fibroblasts (CRL-1474) were obtained from the American Type Culture Collection, ATCC, and cultured using Dulbecco’s modified Eagle’s medium (DMEM) supplemented with 10% fetal bovine serum, 50 U/mL penicillin, and 50 μg/mL streptomycin under a humidified atmosphere [5% CO_2_ at 37 °C] and sterile conditions, including plastics and cell culture reagents purchased from Gibco (Grand Island, NY, USA). When the fibroblasts (passage no. 9) reached 90% confluence, experimental cell groups were prepared as described in Table 1. The conditions of the cell exposure to UVA were selected at 13 J/cm^3^ according to the 70% cell viability appointed using the 3-[4,5-dimethylthiazol-2-yl]-2,5 diphenyl tetrazolium bromide (MTT) assay [69]. Lipid extracts from *N. oceanica* or *C. amblystomatis* were used in the concentration selected at 2 µg/mL, which was not toxic for the cultured skin fibroblasts.

Following the preparation of experimental fibroblast groups as summarized in Table 1, they were scraped from the plates on ice and suspended in Tris-buffered saline (TBS) containing a protease inhibitor cocktail (Sigma-Aldrich, P8340, pH 8.0). Cells were lysed by sonication on ice and centrifuged (15 min, 12,000× *g*) for the downstream analysis. The total protein level of the samples was measured by performing the Bradford assay [70].

### 4.4. Immunoprecipitation Against Human Caspase-1 

The samples containing 100 µg of proteins were used for the analysis of the fibroblast’s proteomic profile in the caspase-1 interactome area through immunoprecipitation (IP) against human caspase-1 antibody. Samples were pre-cleaned with protein A-agarose to remove nonspecifically bound molecules. Protein A-agarose was removed by centrifugation (10,000× *g*, 1 min, 4 °C) and the primary antibody against human caspase-1 (Abcam, Anti-Caspase1, ab1872) was added. Then the samples were incubated for 1h at 4 °C. For precipitating the proteins bound with antibodies, protein A-agarose was added and incubated (overnight). Later, the incubated samples were centrifuged (10,000× *g*, 10 min, 4 °C) and the obtained pellet, as proteins immunoprecipitated with caspase-1, was used for the in-solution digestion step regarding downstream proteomic analysis. 

### 4.5. Preparation of Peptide Mixtures 

For the whole proteome analysis of fibroblasts, samples containing 30 μg of protein was mixed with the sample loading buffer (Laemmli buffer containing 5% 2-mercaptoethanol) in a volume ratio of 1:2 and heated at 100 °C for 7 min to separate proteins in 10% Tris-Glycine SDS-PAGE gels. After electrophoresis, the gels were fixed in methanol/acetic acid/water (4:1:5; for 1 h) and stained with Coomassie Brilliant Blue R-250, overnight. Entire lanes were cut from the gels and sliced into eight sections (Appendix A). Then, reduction and alkylation of the proteins in each slice were performed via incubation with 10 mM 1,4-dithiothreitol (DTT) and 50 mM iodoacetamide (IAA), respectively. They were incubated with trypsin (Promega, Madison, WI, USA) at 37 °C, overnight, for in-gel protein digestion. The obtained peptide mixtures were extracted from the gel, and the dried peptide mixtures dissolved in 5% acetonitrile (ACN) with 0.1% formic acid (FA).

In parallel, the immunoprecipitated samples (with caspase-1) were denatured by mixing with 8 M urea, reduced with 10 mM DTT, and alkylated with 50 mM IAA incubation. And 10 mM DTT was again added to stop the alkylation. Directly after four-fold dilution, the samples were digested (in solution) with trypsin in a ratio of 1:50 (trypsin/proteins), at 37 °C, overnight. And, by ensuring a final concentration in the samples of 0.1%, FA at 10% concentration was added in the samples to stop trypsinization [71]. The final obtained peptide mixture was dried (under inert gas) and dissolved in 5% ACN with 0.1% FA.

### 4.6. Separation of Peptides Using Nano-HPLC

The Ultimate 3000 high-performance liquid chromatography (HPLC) system (Dionex, Idstein, Germany) on a 150 mm × 0.075 mm PepMap RSLC capillary analytical C18 column with a 2 μm particle size (Dionex, LC Packings) was used to separate the final peptide mixtures at a flow rate of 0.300 μL/min. Eluent A (5% ACN + 0.1% FA) and eluent B (90% ACN + 0.03% FA) were applied for the sample mobilization through the column, over a time gradient (starting at 3 min and increasing to 60% eluent B for 40 min). And the peptides were analyzed using a Q Exactive HF mass spectrometer (with positive ion calibration and in data-dependent mode) with an electrospray ionization source (ESI) (Thermo Fisher Scientific, Bremen, Germany). The complete arrangement for liquid chromatography–tandem mass spectrometry (LC-MS/MS) for analysis has been described in detail previously [72].

### 4.7. Identification, Grouping, and Label-Free Quantification of Proteins 

The MaxQuant proteomics software (version 2.5.2.0, search engine Andromeda) [73] was used for the analysis of the raw data generated by LC-MS/MS. The default parameters were used for protein identification by adding asparagine or glutamine deamidation as a variable modification in addition to default methionine oxidation, protein N-term acetylation, and cysteine carbamidomethylation as a fixed modification. Label-free quantification (LFQ) was performed according to the signal intensities of the precursor ions for the semi-quantitative analysis. The data were searched against the UniProtKB-SwissProt database (taxonomy: *Homo sapiens*). 

### 4.8. Statistical Analysis 

Samples from each experimental fibroblast group mentioned above were analyzed in three independent experiments. Only the identified proteins (obtained by the MaxQuant software, version 2.6.2.0.) with at least 2 unique peptides and q-value ≤ 0.005, false discovery rate (FDR) confidence (peptide and protein FDR of 1%), were considered for statistical analysis. The individual protein intensities from LFQ were log-transformed and normalized by the median using the open-source software MetaboAnalyst (version 6.0) [74]. Standard statistical analysis including one-way ANOVA, principal component analysis (PCA), dendrogram, and volcano plot assessments were applied using MetaboAnalyst. ANOVA *p*-value (FDR) cut-off < 0.05 was considered statistically significant. Heatmaps were generated using MetaboAnalyst software. Protein annotations were completed using the STRING (version 12.0) [75] and Perseus (version 2.0.11.0) [76] software platforms, comparatively. Protein profile plots were generated by Perseus using average protein intensities (log-transformed and normalized by median) for each cell group, as explained in Table 1. The heatmap of the Pearson correlation matrix was generated by MetaboAnalyst software. And the network for protein–protein interactions was generated by the Cytoscape software version v3.10.2 using stringApp version 2.1.1 [77,78] and default settings (confidence: 0.400; FDR stringency: 5 percent). 

## 5. Conclusions

The obtained proteomic data suggest the cytoprotective activity of both *N. oceanica* and *C. amblystomatis* lipid extracts in skin fibroblasts, by restoring the control expression of antioxidant proteins and proteins involved in inflammatory signaling, which are both impaired by the effect of UVA irradiation. The modulation of the interplay between the intracellular redox state and inflammatory signaling appears to stem from the cytoprotective effects of the algal lipid extracts against the UVA-mediated oxidative stress. These extracts interfere with the regulation of HO-1 and the TRX expression, which seems to be a critical molecular mechanism in combating the UVA-induced oxidative imbalance. It has been observed that the *C. amblystomatis* lipid extract is more effective in reversing the expression of antioxidant/cytoprotective proteins suppressed by UVA. However, both algal lipid extracts show the potential to change inflammatory signaling. Furthermore, it should be noted that both lipid extracts significantly change the protein profile in terms of the redox regulation and inflammatory signaling in control fibroblasts.

The cytoprotective effects of the *N. oceanica* and *C. amblystomatis* lipid extracts in metabolic disorders of skin fibroblasts caused by UVA radiation suggest a promising new therapeutic approach that harnesses natural resources to protect skin cells from the harmful effects of UV radiation and, consequently, to mitigate the development of skin diseases.

## Figures and Tables

**Figure 1 marinedrugs-22-00509-f001:**
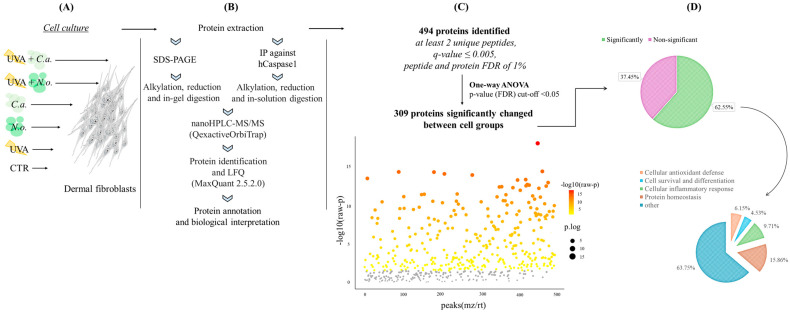
Summary of the experimental design presenting the experimental cell groups [preparation of the **CTR**, ***N.o.***, ***C.a.***, **UVA**, **UVA + *N.o.***, and **UVA + *C.a.*** cell groups, as explained in Table 1] (**A**), proteomic sample preparation including SDS-PAGE or IP (immunoprecipitation) and MS/MS proteomic analysis (**B**), as well as the obtained complete proteomic data with the main statistical thresholds used (**C**) and a general assessment of the obtained protein groups highlighting their biological functions (cellular antioxidant defense/cell survival and differentiation/cellular inflammatory response/protein homeostasis), and the percentage of identified protein numbers (for each protein group) compared to the number of total identified proteins (**D**). “Significant” is used for the proteins whose intensities are statistically significantly changed between experimental cell groups according to the one-way ANOVA. “Non-significant” is used for the proteins whose intensities are not changed statistically significantly between cell groups according to the one-way ANOVA. (hCaspase1, human caspase-1; LFQ, label-free quantification; FDR, false discovery rate; nanoHPLC, nano-high-performance liquid chromatography; MS, mass spectrometry; QexactiveOrbiTrap, Q Exactive HF mass spectrometer.)

**Figure 2 marinedrugs-22-00509-f002:**
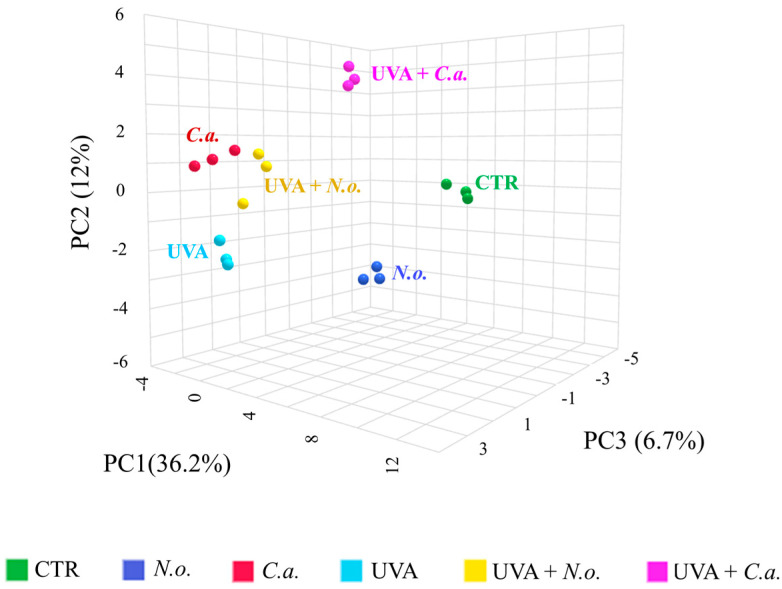
Principal component analysis (PCA) of the skin fibroblasts’ proteome from the experimental cell groups **CTR**, ***N.o.***, ***C.a.***, **UVA**, **UVA + *N.o.***, and **UVA + *C.a.*** showing an obvious clustering of CTR and UVA samples in distinct groups. (**CTR**, control cells cultured in the standard growing medium; ***N.o.***, cells cultured in the standard growing medium also containing lipid extracts from *N. oceanica* (2 µg/mL) for 24 h; **C.a.**, cells cultured in the standard growing medium containing lipid extracts from *C. amblystomatis* (2 µg/mL) for 24 h; **UVA**, cells exposed to UVA (365 nm) at a dose of 13 J/cm^3^ and then incubated in the standard growing medium for 24 h; **UVA + *N.o.***, cells exposed to UVA (365 nm) at a dose of 13 J/cm^3^ and then incubated in the standard growing medium containing lipid extracts from *N. oceanica* (2 µg/mL) for 24 h; **UVA + *C.a.***, cells exposed to UVA (365 nm) at a dose of 13 J/cm^3^ and then incubated in the standard growing medium containing lipid extracts from *C. amblystomatis* (2 µg/mL) for 24 h; **PC1/2/3**, principal component 1/2/3.)

**Figure 3 marinedrugs-22-00509-f003:**
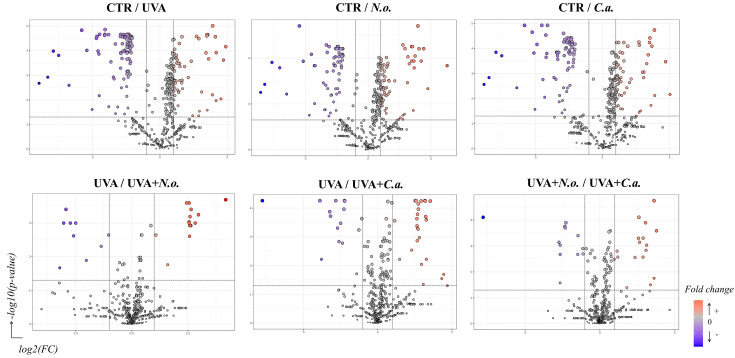
Volcano plots comparing the effects of *N. oceanica* and *C. amblystomatis* lipid extracts on the proteome of skin fibroblasts from the experimental cell groups (**CTR**, control cells cultured in the standard growing medium; ***N.o.***, cells cultured in the standard growing medium also containing lipid extracts from *N. oceanica* (2 µg/mL) for 24 h; ***C.a.***, cells cultured in the standard growing medium containing lipid extracts from *C. amblystomatis* (2 µg/mL) for 24 h; **UVA**, cells exposed to UVA (365 nm) at a dose of 13 J/cm^3^ and then incubated in the standard growing medium for 24 h; **UVA + *N.o.***, cells exposed to UVA (365 nm) at a dose of 13 J/cm^3^ and then incubated in the standard growing medium containing lipid extracts from *N. oceanica* (2 µg/mL) for 24 h; **UVA + *C.a.***, cells exposed to UVA (365 nm) at a dose of 13 J/cm^3^ and then incubated in the standard growing medium containing lipid extracts from *C. amblystomatis* (2 µg/mL) for 24 h). [Significant features (in blue and red as seen in figure—fold change) had *p* < 0.05].

**Figure 4 marinedrugs-22-00509-f004:**
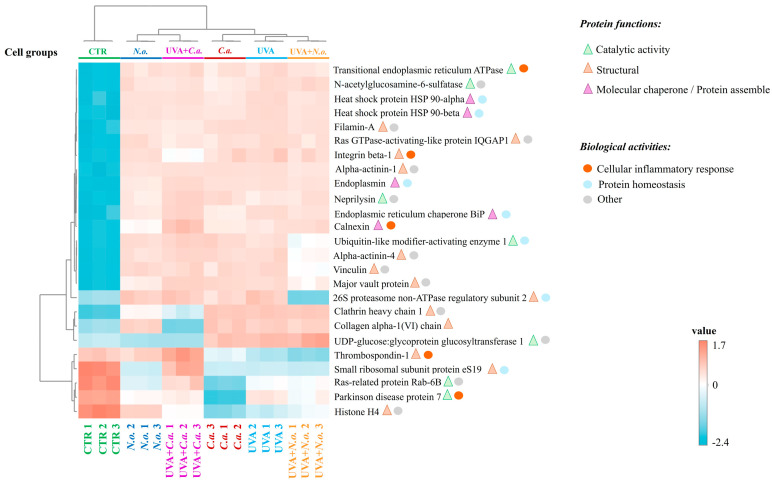
Heatmap, clustering of the top 25 proteins with significantly changed expressions in the skin fibroblasts of the following cell groups: control group (**CTR**), *N. oceanica* or *C. amblystomatis* lipid extract-treated group (***N.o.***/***C.a.***), UVA-irradiated cell group (**UVA**), and cell group treated with *N. oceanica* or *C. amblystomatis* lipid extracts following UVA irradiation (**UVA + *N.o.***/**UVA + *C.a.***), as explained in Table 1. While the protein functions of the mentioned proteins were included, the proteins involved in the regulation of inflammation or protein homeostasis were also indicated.

**Figure 5 marinedrugs-22-00509-f005:**
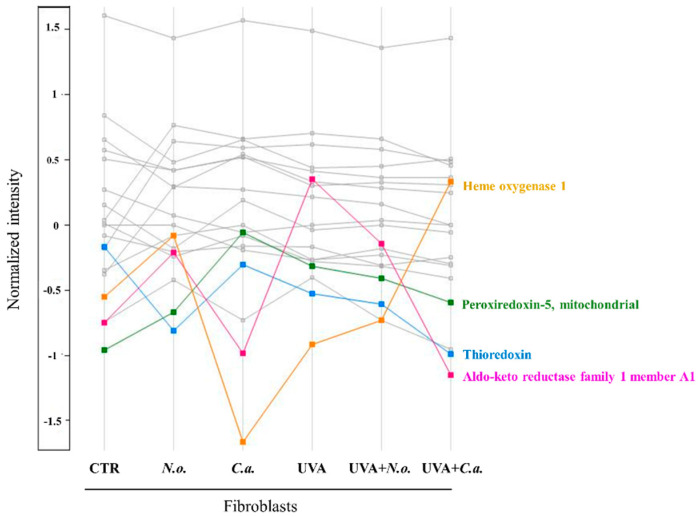
Profile plot generated by the Perseus software platform (version 2.0.11.0), using the average intensities (log-transformed and normalized by median) of proteins with cytoprotective activity against oxidative stress whose levels were significantly changed between cell groups (**CTR**, control cells cultured in the standard growing medium; ***N.o.***, cells cultured in the standard growing medium also containing lipid extracts from *N. oceanica* (2 µg/mL) for 24 h; ***C.a.***, cells cultured in the standard growing medium containing lipid extracts from *C. amblystomatis* (2 µg/mL) for 24 h; **UVA**, cells exposed to UVA (365 nm) at a dose of 13 J/cm^3^ and then incubated in the standard growing medium for 24 h; **UVA + *N.o.***, cells exposed to UVA (365 nm) at a dose of 13 J/cm^3^ and then incubated in the standard growing medium containing lipid extracts from *N. oceanica* (2 µg/mL) for 24 h; **UVA + *C.a.***, cells exposed to UVA (365 nm) at a dose of 13 J/cm^3^ and then incubated in the standard growing medium containing lipid extracts from *C. amblystomatis* (2 µg/mL) for 24 h). Only the proteins showing at least 3-fold changes (log_2_FC) were highlighted with pink, blue, green, and yellow colors (different than grey color). Except for thioredoxin, *N. oceanica* or *C. amblystomatis* lipid extracts were able to reverse UVA-disrupted expression of aldo-keto reductase family member A1, peroxiredoxin-5, mitochondrial, and heme oxygenase-1 towards their CTR-level expressions. Statistics for the indicated proteins (mean values ± SD of three independent samples and statistically significant differences for *p* ≤ 0.05) are presented in Appendix A.

**Figure 6 marinedrugs-22-00509-f006:**
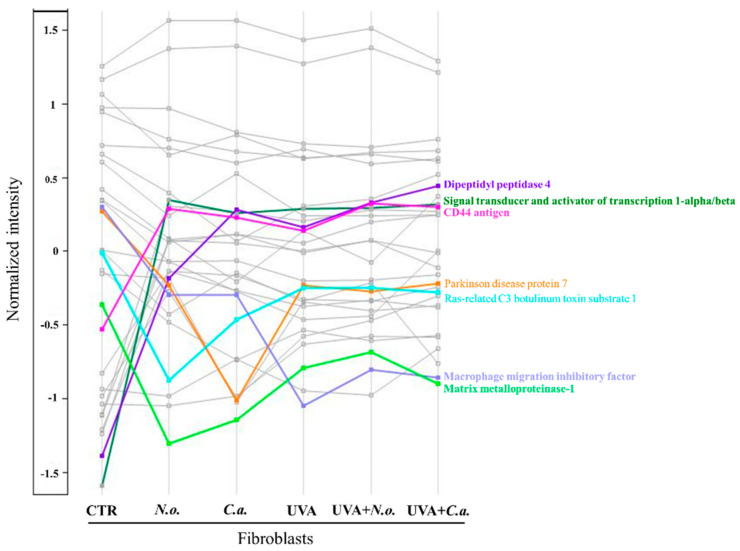
Profile plot generated by the Perseus software platform (version 2.0.11.0), using the average intensities (log-transformed and normalized by median) of proteins participating in the regulation of cellular inflammatory response whose levels were significantly changed between cell groups (**CTR**, control cells cultured in the standard growing medium; ***N.o.***, cells cultured in the standard growing medium also containing lipid extracts from *N. oceanica* (2 µg/mL) for 24 h; ***C.a.***, cells cultured in the standard growing medium containing lipid extracts from *C. amblystomatis* (2 µg/mL) for 24 h; **UVA**, cells exposed to UVA (365 nm) at a dose of 13 J/cm^3^ and then incubated in the standard growing medium for 24 h; **UVA + *N.o.***, cells exposed to UVA (365 nm) at a dose of 13 J/cm^3^ and then incubated in the standard growing medium containing lipid extracts from *N. oceanica* (2 µg/mL) for 24 h; **UVA + *C.a.***, cells exposed to UVA (365 nm) at a dose of 13 J/cm^3^ and then incubated in the standard growing medium containing lipid extracts from *C. amblystomatis* (2 µg/mL) for 24 h). Only the proteins showing at least 3-fold changes (log_2_FC) were highlighted with different colors (different than grey). Statistics for the indicated proteins (mean values ± SD of three independent samples and statistically significant differences for *p* ≤ 0.05) are presented in Appendix A.

**Figure 7 marinedrugs-22-00509-f007:**
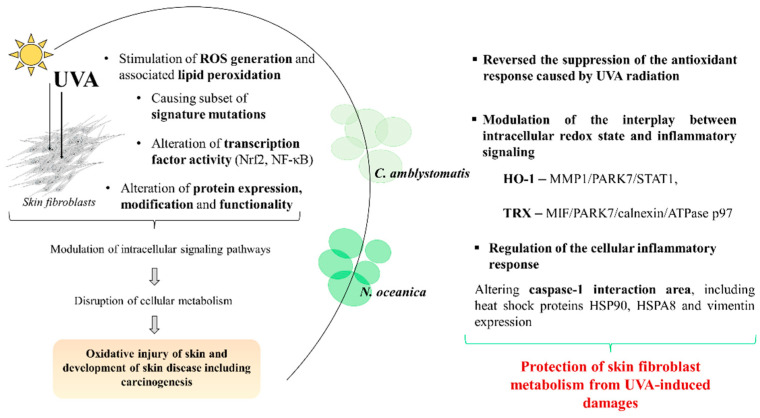
*N. oceanica* and *C. amblystomatis* lipid extracts restore the UVA-impaired control expression of antioxidant proteins and proteins involving inflammatory signaling in skin fibroblasts (ultraviolet A, UVA; reactive oxygen species, ROS; nuclear factor erythroid 2-related factor 2, Nrf2; nuclear factor kappa, NF-κB; heme oxygenase-1, HO-1; aldo-keto reductase family member 1 A1, AKR1A1; thioredoxin, Trx; matrix metalloproteinase-1, MMP1; Parkinson’s disease protein 7, PARK7; signal transducer and activator of transcription1-alpha/beta, STAT1; macrophage migration inhibitory factor, MIF).

**Table 1 marinedrugs-22-00509-t001:** Experimental groups of fibroblasts used in the downstream proteomic analysis and the methods used for their preparation. All experimental groups were prepared as three biological replicates.

Experimental Fibroblast Group	Group Name	Method Used for Preparation
Control group	**CTR**	Fibroblasts were cultured with the standard growing medium.
Cells treated with lipid extract from *Nannochloropsis oceanica* (*N.o.*)	** *N.o.* **	Fibroblasts were cultured with the standard growing medium also containing the lipid extracts from *N. oceanica* (2 µg/mL) for 24 h.
Cells treated with lipid extract from *Chlorococcum amblystomatis* (*C.a.*)	** *C.a.* **	Fibroblasts were cultured with the standard growing medium containing the lipid extracts from *C. amblystomatis* (2 µg/mL) for 24 h.
Cells exposed to UVA radiation	**UVA**	Fibroblasts were exposed to UVA (365 nm) at a dose of 13 J/cm^3^ and then incubated with the standard growing medium for 24 h.
Cells exposed to UVA radiation and treated with lipid extract from *Nannochloropsis oceanica*	**UVA + *N.o.***	Fibroblasts were exposed to UVA (365 nm) at a dose of 13 J/cm^3^ and then incubated with the standard growing medium containing lipid extracts from *N. oceanica* (2 µg/mL) for 24 h.
Cells exposed to UVA radiation and treated with lipid extract from *Chlorococcum amblystomatis*	**UVA + *C.a.***	Fibroblasts were exposed to UVA (365 nm) at a dose of 13 J/cm^3^ and then incubated with the standard growing medium containing lipid extracts from *C. amblystomatis* (2 µg/mL) for 24 h.

**Table 2 marinedrugs-22-00509-t002:** Fold changes between cell groups (**CTR, *N.o.***, ***C.a.***, **UVA**, **UVA + *N.o.****,* and **UVA + *C.a***., as explained Table 1), protein biological roles, and UNIPROT IDs. “NA” used for “not applicable, depending on the non-detection of associated protein intensity”. In log2(FC)segment, “decrease” is highlighted with red color and “increase” is highlighted with green color, as visible with high to low color intensity associated with their fold change proportions.

			Fold Changes									
			UVA/CTR		No/CTR		Ca/CTR		UVA + *N.o*./UVA		UVA + *C.a*./UVA	
Biological Role	ID Protein Name		Fold Change	log_2_(FC)	Fold Change	log_2_(FC)	Fold Change	log_2_(FC)	Fold Change	log_2_(FC)	Fold Change	log_2_(FC)
Cellular defense against oxidative stress	P14550 Aldo-keto reductase family 1 member A1		9.71	3.28	2.43	1.28	0.20	−2.29	0.31	−1.68	0.05	−4.41
	Q04828 Aldo-keto reductase family 1 member C1		2.60	1.38	2.87	1.52	2.33	1.22	0.86	−0.22	0.66	−0.59
	P15121 Aldo-keto reductase family 1 member B1		4.22	2.08	4.41	2.14	3.01	1.59	0.88	−0.18	0.61	−0.72
	P30048 Thioredoxin-dependent peroxide reductase, mitochondrial		0.72	−0.47	0.62	−0.69	0.58	−0.79	0.70	−0.51	0.90	−0.15
	P10599 Thioredoxin		0.42	−1.26	0.10	−3.37	0.51	−0.96	0.82	−0.28	0.48	−1.05
	Q16881 Thioredoxin reductase 1, cytoplasmic		0.48	−1.06	0.83	−0.28	0.45	−1.14	1.06	0.09	0.99	−0.01
	P15559 NAD(P)H dehydrogenase [quinone] 1		0.47	−1.09	0.53	−0.93	0.33	−1.61	1.07	0.10	1.08	0.12
Cellular defense against oxidative stress	Q08257 Quinone oxidoreductase		4.49	2.17	4.25	2.09	**NA**	**NA**	0.16	−2.67	0.16	−2.61
	P32119 Peroxiredoxin-2		0.45	−1.16	0.46	−1.13	0.56	−0.83	0.90	−0.16	0.81	−0.30
	P30044 Peroxiredoxin-5, mitochondrial		6.81	2.77	3.15	1.66	9.84	3.30	0.80	−0.33	0.61	−0.71
	Q13162 Peroxiredoxin-4		0.71	−0.49	0.67	−0.57	0.72	−0.48	0.88	−0.18	0.97	−0.04
	Q06830 Peroxiredoxin-1		0.35	−1.52	0.36	−1.46	0.45	−1.14	1.01	0.02	1.28	0.35
	P30041 Peroxiredoxin-6		0.58	−0.80	0.38	−1.39	0.77	−0.38	1.05	0.08	1.02	0.03
	P09601 Heme oxygenase-1		0.03	−4.93	2.44	1.29	0.08	−3.69	**NA**	**NA**	225.77	7.82
	P78417 Glutathione S-transferase omega-1		0.47	−1.08	0.58	−0.79	0.63	−0.68	1.03	0.04	1.09	0.13
	P09211 Glutathione S-transferase P		0.42	−1.24	0.36	−1.49	0.54	−0.89	0.87	−0.21	0.89	−0.17
	P16152 Carbonyl reductase [NADPH] 1		1.07	0.09	1.52	0.60	1.55	0.63	1.18	0.24	1.01	0.01
	P00387 NADH-cytochrome b5 reductase 3		5.87	2.55	5.51	2.46	4.28	2.10	0.89	−0.17	0.79	−0.34
	P04179 Superoxide dismutase [Mn], mitochondrial		0.68	−0.57	0.56	−0.85	0.64	−0.64	0.72	−0.47	0.95	−0.07
Cell survival and differentiation	Q6UVY6 DBH-like monooxygenase protein 1		5.14	2.36	5.31	2.41	4.19	2.07	0.83	−0.26	0.94	−0.09
	P13473 Lysosome-associated membrane glycoprotein 2		3.84	1.94	4.45	2.15	4.49	2.17	1.27	0.34	1.16	0.22
	Q99829 Copine-1		0.88	−0.19	1.24	0.31	0.12	−3.05	0.42	−1.26	0.13	−2.92
	Q8WUM4 Programmed cell death 6-interacting protein		5.70	2.51	4.34	2.12	3.13	1.64	1.11	0.15	0.95	−0.08
	Q96TA1 Protein Niban 2		1.16	0.21	3.90	1.96	0.29	−1.79	0.55	−0.87	1.52	0.60
	P07384 Calpain-1 catalytic subunit		6.21	2.63	5.03	2.33	5.11	2.35	1.02	0.02	1.20	0.27
	Q12884 Prolyl endopeptidase FAP		25.17	4.65	3.39	1.76	20.64	4.37	20.64	4.37	1.38	0.46
	Q9ULV4 Coronin-1C		0.60	−0.73	0.57	−0.82	0.53	−0.91	0.94	−0.08	1.02	0.03
	P17655 Calpain-2 catalytic subunit		4.47	2.16	5.90	2.56	4.60	2.20	1.08	0.12	0.94	−0.09
	P37802 Transgelin-2		0.54	−0.88	0.49	−1.04	0.79	−0.35	0.94	−0.09	0.89	−0.17
	P99999 Cytochrome c		0.72	−0.48	0.59	−0.76	0.70	−0.51	0.89	−0.17	1.37	0.46
	P21796 Voltage-dependent anion-selective channel protein 1		0.68	−0.55	0.95	−0.07	0.62	−0.70	1.02	0.03	1.47	0.55
	P45880 Voltage-dependent anion-selective channel protein 2		3.47	1.80	3.87	1.95	1.64	0.72	0.76	−0.40	0.99	−0.02
	P62937 Peptidyl-prolyl cis-trans isomerase A		0.53	−0.92	0.42	−1.24	0.70	−0.52	0.98	−0.03	0.81	−0.31
Cellular inflammatory response	P16070 CD44 antigen		6.93	2.79	8.08	3.02	7.07	2.82	1.20	0.26	1.24	0.32
	P17858 ATP-dependent 6-phosphofructokinase, liver type		3.92	1.97	**NA**	**NA**	4.88	2.29	0.73	−0.45	1.51	0.60
	Q14956 Transmembrane glycoprotein NMB		1.41	0.50	1.17	0.23	0.94	−0.09	0.62	−0.68	0.74	−0.43
	P07996 Thrombospondin-1		0.50	−1.00	1.02	0.03	0.65	−0.62	0.83	−0.27	2.53	1.34
	Q12905 Interleukin enhancer-binding factor 2		0.58	−0.78	0.55	−0.88	0.30	−1.72	0.68	−0.56	0.80	−0.32
	P53396 ATP-citrate synthase		4.75	2.25	6.69	2.74	5.02	2.33	0.85	−0.23	0.84	−0.26
	P20591 Interferon-induced GTP-binding protein Mx1		2.65	1.41	1.09	0.12	0.32	−1.63	0.89	−0.16	0.96	−0.05
	P03956 Interstitial collagenase/matrix metalloproteinase-1		0.04	−4.75	0.16	−2.67	0.04	−4.76	15.66	3.97	10.51	3.39
Cellular inflammatory response	P62834 Ras-related protein Rap-1A		0.43	−1.21	0.48	−1.07	0.49	−1.03	0.79	−0.33	0.94	−0.09
	P05556 Integrin beta-1		5.81	2.54	5.59	2.48	5.48	2.45	0.91	−0.13	0.65	−0.61
	P14174 Macrophage migration inhibitory factor		0.09	−3.54	0.35	−1.51	0.37	−1.42	**NA**	**NA**	**NA**	**NA**
	Q99623 Prohibitin-2		2.85	1.51	4.55	2.19	0.34	−1.56	0.99	−0.01	1.61	0.69
	P27824 Calnexin		4.94	2.31	4.32	2.11	4.72	2.24	1.09	0.12	1.34	0.42
	P63000 Ras-related C3 botulinum toxin substrate 1		0.87	−0.21	0.12	−3.08	0.48	−1.06	0.79	−0.35	0.80	−0.33
	P19367 Hexokinase-1		5.14	2.36	6.48	2.70	6.17	2.62	0.76	−0.40	1.04	0.05
	P60903 Protein S100-A10		0.78	−0.36	0.56	−0.83	0.32	−1.65	0.48	−1.07	1.48	0.56
	Q9BTV4 Transmembrane protein 43		0.99	−0.01	1.16	0.21	0.18	−2.46	1.04	0.06	0.36	−1.49
	P31949 Protein S100-A11		0.67	−0.57	0.46	−1.13	0.88	−0.19	0.68	−0.55	0.81	−0.30
	P07339 Cathepsin D		0.73	−0.45	0.80	−0.32	0.67	−0.58	0.83	−0.27	0.81	−0.30
	P26447 Protein S100-A4		0.65	−0.62	0.53	−0.92	1.04	0.06	0.79	−0.35	0.86	−0.21
Cellular inflammatory response	P17931 Galectin-3		0.66	−0.60	0.64	−0.65	0.73	−0.45	0.94	−0.08	0.85	−0.23
	Q99497 Parkinson’s disease protein 7		0.47	−1.09	0.39	−1.37	0.07	−3.93	0.72	−0.47	0.89	−0.17
	P07858 Cathepsin B		0.85	−0.24	1.21	0.27	0.84	−0.25	0.75	−0.42	0.92	−0.12
	P27487 Dipeptidyl peptidase 4		16.37	4.03	6.02	2.59	17.80	4.15	1.15	0.20	1.64	0.71
	P42224 Signal transducer and activator of transcription 1-alpha/beta		101.95	6.67	96.47	6.59	79.05	6.30	0.80	−0.33	0.91	−0.13
	P55072 Transitional endoplasmic reticulum ATPase		4.30	2.11	4.43	2.15	4.14	2.05	0.94	−0.09	1.00	−0.003
	Q03135 Caveolin-1		0.66	−0.60	0.66	−0.60	0.32	−1.66	0.88	−0.19	1.40	0.49
	P09382 Galectin-1		0.54	−0.88	0.48	−1.07	0.66	−0.60	0.87	−0.20	0.98	−0.03
	P04083 Annexin A1		1.91	0.94	1.97	0.98	2.10	1.07	1.00	0.01	0.75	−0.42
Cellular inflammatory response	P04406 Glyceraldehyde-3-phosphate dehydrogenase		2.32	1.22	2.53	1.34	2.59	1.37	0.93	−0.11	0.61	−0.71
Protein homeostasis	Q13619 Cullin-4A		5.29	2.40	9.16	3.20	7.14	2.84	0.97	−0.04	1.08	0.12
	Q96AY3 Peptidyl-prolyl cis-trans isomerase FKBP10		3.76	1.91	4.36	2.12	4.52	2.18	1.05	0.07	1.08	0.10
	P36578 Large ribosomal subunit protein uL4		0.55	−0.85	0.58	−0.78	0.69	−0.54	1.02	0.03	1.12	0.17
	Q13200 26S proteasome non-ATPase regulatory subunit 2		5.26	2.39	5.32	2.41	4.22	2.08	0.13	−2.90	0.99	−0.01
	Q86VP6 Cullin-associated NEDD8-dissociated protein 1		7.35	2.88	9.96	3.32	10.23	3.35	0.89	−0.17	0.78	−0.36
	P15880 Small ribosomal subunit protein uS5		0.18	−2.50	0.08	−3.64	0.07	−3.74	0.39	−1.36	1.07	0.09
	O95302 Peptidyl-prolyl cis-trans isomerase FKBP9		0.39	−1.38	2.91	1.54	2.76	1.47	6.89	2.79	6.64	2.73
	P34932 Heat shock 70 kDa protein 4		5.15	2.36	5.32	2.41	4.86	2.28	0.72	−0.47	1.01	0.01
Protein homeostasis	P62906 Large ribosomal subunit protein uL1		0.08	−3.59	0.42	−1.25	0.08	−3.60	**NA**	**NA**	3.67	1.88
	P25789 Proteasome subunit alpha type-4		0.51	−0.96	0.58	−0.78	0.48	−1.05	1.00	0.00	0.88	−0.18
	P62424 Large ribosomal subunit protein eL8		0.19	−2.38	0.42	−1.25	0.22	−2.19	1.19	0.25	1.28	0.36
	P39019 Small ribosomal subunit protein eS19		0.09	−3.46	0.10	−3.38	0.09	−3.46	**NA**	**NA**	6.43	2.69
	P62249 Small ribosomal subunit protein uS9		0.37	−1.42	0.33	−1.59	0.49	−1.03	1.00	0.00	1.41	0.50
	P38646 Stress-70 protein, mitochondrial		21.26	4.41	20.31	4.34	18.37	4.20	0.99	−0.01	1.07	0.10
	P62269 Small ribosomal subunit protein uS13		0.42	−1.25	0.54	−0.89	0.74	−0.44	0.82	−0.28	1.61	0.69
	Q07020 Large ribosomal subunit protein eL18		0.35	−1.52	0.83	−0.27	0.37	−1.43	1.24	0.31	0.89	−0.18
	Q96FW1 Ubiquitin thioesterase OTUB1		2.84	1.51	3.21	1.68	1.15	0.20	0.59	−0.76	0.12	−3.12
	Q58FF8 Putative heat shock protein HSP 90-beta 2		5.17	2.37	5.61	2.49	4.47	2.16	0.96	−0.05	0.77	−0.38
	P05388 Large ribosomal subunit protein uL10		0.80	−0.33	1.09	0.12	0.90	−0.16	1.08	0.11	0.88	−0.18
	P46782 Small ribosomal subunit protein uS7		0.31	−1.71	0.22	−2.17	0.55	−0.86	1.04	0.05	1.41	0.49
	P62081 Small ribosomal subunit protein eS7		4.90	2.29	NA	NA	6.90	2.79	1.14	0.18	1.39	0.47
Protein homeostasis	P62277 Small ribosomal subunit protein uS15		0.06	−3.96	0.07	−3.87	0.41	−1.28	3.35	1.74	6.81	2.77
	P46781 Small ribosomal subunit protein uS4		0.06	−4.17	0.36	−1.48	0.13	−2.89	2.11	1.08	5.47	2.45
	Q06323 Proteasome activator complex subunit 1		0.49	−1.02	0.57	−0.82	0.49	−1.02	0.85	−0.24	0.70	−0.51
	P30050 Large ribosomal subunit protein uL11		0.49	−1.03	0.55	−0.86	0.67	−0.57	0.96	−0.05	1.19	0.25
	P60900 Proteasome subunit alpha type-6		0.50	−1.00	0.59	−0.76	0.55	−0.86	0.95	−0.07	1.17	0.23
	P25787 Proteasome subunit alpha type-2		0.35	−1.51	0.16	−2.66	0.44	−1.18	0.43	−1.23	1.31	0.39
	O14818 Proteasome subunit alpha type-7		0.59	−0.76	0.65	−0.61	0.57	−0.80	1.00	−0.01	0.84	−0.25
	P62241 Small ribosomal subunit protein eS8		0.05	−4.22	0.06	−4.13	0.28	−1.85	2.13	1.09	9.81	3.29
	P0DMV9 Heat shock 70 kDa protein 1B		1.14	0.19	5.24	2.39	6.45	2.69	7.75	2.95	5.43	2.44
	P08238 Heat shock protein HSP 90-beta		41.47	5.37	38.24	5.26	33.23	5.05	0.84	−0.25	0.82	−0.28
	P63244 Small ribosomal subunit protein RACK1		0.31	−1.69	0.70	−0.51	0.44	−1.18	1.25	0.32	1.64	0.71
	P23396 Small ribosomal subunit protein uS3		0.11	−3.15	0.01	−6.19	0.03	−5.09	1.05	0.07	3.07	1.62
	Q9UL46 Proteasome activator complex subunit 2		0.45	−1.15	0.56	−0.84	0.50	−1.00	1.01	0.02	0.80	−0.32
Protein homeostasis	P22314 Ubiquitin-like modifier-activating enzyme 1		5.27	2.40	5.79	2.53	5.85	2.55	0.72	−0.47	1.12	0.17
	P07900 Heat shock protein HSP 90-alpha		22.31	4.48	20.72	4.37	17.61	4.14	0.84	−0.26	0.89	−0.18
	Q15084 Protein disulfide-isomerase A6		0.55	−0.86	0.64	−0.65	0.53	−0.91	0.92	−0.13	0.96	−0.05
	P11142 Heat shock cognate 71 kDa protein		1.86	0.90	3.43	1.78	3.23	1.69	1.94	0.96	1.63	0.71
	P08865 Small ribosomal subunit protein uS2		0.51	−0.98	0.73	−0.46	0.64	−0.65	0.88	−0.19	1.29	0.37
	P11021 Endoplasmic reticulum chaperone BiP		23.10	4.53	19.80	4.31	16.83	4.07	0.88	−0.18	1.12	0.17
	P27348 14-3-3 protein theta		0.31	−1.68	0.39	−1.35	0.53	−0.92	1.44	0.53	1.24	0.31
	P04792 Heat shock protein beta-1		0.50	−0.99	0.59	−0.76	0.65	−0.63	1.01	0.02	0.92	−0.13
	P62258 14-3-3 protein epsilon		0.49	−1.03	0.71	−0.50	0.47	−1.09	1.04	0.06	0.90	−0.15
	P63104 14-3-3 protein zeta/delta		0.47	−1.09	0.56	−0.85	0.54	−0.88	1.02	0.03	0.89	−0.16
	P23284 Peptidyl-prolyl cis-trans isomerase B		0.64	−0.65	0.53	−0.93	0.61	−0.71	0.90	−0.15	1.04	0.06
	P0CG47 Polyubiquitin-B		0.41	−1.27	0.53	−0.91	0.49	−1.04	0.80	−0.33	1.95	0.97
Protein homeostasis	P30101 Protein disulfide-isomerase A3		0.67	−0.57	0.67	−0.57	0.61	−0.72	0.83	−0.27	0.83	−0.27
	P07237 Protein disulfide-isomerase		0.53	−0.91	0.67	−0.58	0.49	−1.02	0.96	−0.05	1.07	0.10
	P62701 Small ribosomal subunit protein eS4, X isoform		0.23	−2.15	0.32	−1.63	0.47	−1.10	1.28	0.36	1.93	0.95

## Data Availability

The data showing the raw protein intensities obtained from mass spectrometry analysis is shared in Appendix A. And the data (for additional information) will be made available on request.

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
