# Peer review of "Comparison of Microalgae Nannochloropsis oceanica and Chlorococcum amblystomatis Lipid Extracts Effects on UVA-Induced Changes in Human Skin Fibroblasts Proteome"

_marinedrugs, 2024, doi:10.3390/md22110509_

Round 1
Reviewer 1 Report
Comments and Suggestions for Authors
This was a very interesting study. I have several suggestions for the authors' consideration:
1) Introduction. As this is an in vitro study, it would be worthwhile providing information about the likelihood skin fibroblasts would be exposed to UVA and UVB irradiation. Provide information on the depth of penetration of UVA and UVB solar radiation in the epidermis/dermis of the skin.
2) Line 27. As this was an in vitro study using cultured fibroblasts, care should be given in attributing efficacy in skin.
3) Line 103. Although Table 1 provides details of the experimental groups, this hasn't been introduced yet. Table 1 should be included here.
4) The legend to Figure 1 needs to explain what we are looking at. For example, what is the graph showing?
5) Line 200. It is difficult to interpret what a 0.05-fold change in protein level is describing. I suggest expressing the protein expression as a percentage of the UVA irradiation group.
6) Line 215. The change is either statistically significant, or it is not. Rather than stating "the most significant changes", state "the largest changes".
7) The Results section is very long. Inclusion of Supplementary Table 3 in the main body of the text would help greatly and reduce the amount of text needed to describe the fold changes.
8) The n value is low (n=3). The data should give an indication of the variability between experiments.
9) Figure 5. How do you explain the lower heme oxygenase-1 expression for C.a. compared to UVA? Is C.a. in the absence of UVA harmful?
10) Line 318. Were these pull-down assays? If not, the correlation of change in protein expression does not infer linkage of the proteins (line 324).
11) What is the mechanism for altered regulation of protein expression by the lipids, noting that this is an acute exposure of cells to the lipids?
12) Line 475. This reasoning is not clear. MMP-1 levels were reduced in the absence of the lipid extract.
13) Which fatty acids are present in each algal source? Can you speculate on the fatty acids that are responsible for the observed differences between the two types of lipid extract?
14) Formatting inconsistencies were noted in the Reference section.
- Journals are abbreviated (e.g. ref 6) or not (e.g. ref 13).
- inconsistent page numbering (e.g. ref 11 versus ref 16).
Comments on the Quality of English Language
Grammatical errors were identified throughout the manuscript. These should be corrected.
Use of the word "Moreover" was used excessively.
Line 255. What is "closed results"?
Line 256. These are not temporal changes and so its not appropriate to infer "slowed" rate of expression.
Line 262. Replace "remarkably" with "markedly" and give the fold change.
Line 407. cell cycle progression.
Author Response
Answers for Reviewer 1
Manuscript title: Comparison of the effect of the microalgae Nannochloropsis oceanica and Chlorococcum amblystomatis lipid extracts on the redox-dependent metabolic changes in human skin fibroblasts induced by UVA radiation
Manuscript: Marinedrugs-3293498
-------------------------------------------------------------------------------------------------------------------------------
This was a very interesting study. I have several suggestions for the authors' consideration:
Answer
The authors would like to thank Reviewer for preparing a very precise review of the study. All the reviewer's comments were taken into account during the improvement of the manuscript. We hope that it significantly improved its quality.
Changes to the manuscript were introduced in different colors dedicated to the comments of 3 Reviewers. Changes prepared in response to the Reviewer 1 comments are marked in yellow.
1) Introduction. As this is an in vitro study, it would be worthwhile providing information about the likelihood skin fibroblasts would be exposed to UVA and UVB irradiation. Provide information on the depth of penetration of UVA and UVB solar radiation in the epidermis/dermis of the skin.
Answer
Mentioning the wavelength information, the penetration of UVA and UVB by the different skin layers is included in introduction and the relevant section is improved as follows:
Due to the wavelength characteristics, UV radiation is classified as follows: ultraviolet A (UVA), B (UVB) and C (UVC) with the UV wavelengths of 315–400, 280–315 and 200–280 nm, respectively [3]. However UVC is mostly absorbed by the ozone in the atmosphere [3]. Both UVA and UVB radiation reaching the skin may cause metabolic alterations in skin cells that account for the causes of the development of skin diseases including psoriasis, atopic dermatitis [4] and even skin cancers, including melanoma [5]. While UVB penetrates mainly the epidermis and papillary dermis, UVA affects the dermis, namely its full thickness including the subcutaneous fat [6]. Consequently, photoaging in the dermis is marked with a loss of fibroblast density, extracellular matrix alteration including accumulation of elastin-rich elastic fibres in the reticular dermis [6].
2) Line 27. As this was an in vitro study using cultured fibroblasts, care should be given in attributing efficacy in skin.
Answer
Abstract is improved, by highlighting the therapeutic potential of this study as an in vitro study, as follows:
Thus, those extracts have proven a great potential to be used in cosmetic or cosmeceutical products to protect the skin against the effects of solar radiation. However, the possibility of their use requires evaluation of the effects at the skin level in in vivo and clinical studies.
3) Line 103. Although Table 1 provides details of the experimental groups, this hasn't been introduced yet. Table 1 should be included here.
Answer
The Table 1 introducing the experimental cell groups has been moved at the beginning of the results description. Also the short paragraph-the introductory part of this table has been added as follow:
This study presents the results of the experiment aimed to compare the protective effects of lipid extracts from microalgae N. oceanica and C. amblystomatis against the UVA-induced changes in the proteome of human skin fibroblasts. The compared cell groups with their abbreviations used are described in Table 1.
4) The legend to Figure 1 needs to explain what we are looking at. For example, what is the graph showing?
Answer
The elements of the Figure 1 have been clarified as follows
Figure 1. Summary of the experimental design presenting the experimental cell groups [preparation of the CTR, N.o., C.a., UVA, UVA+N.o. and UVA+C.a. cell groups as explained in Table 1] (A), proteomic sample preparation including SDS-PAGE or IP (immunoprecipitation) and MS/MS proteomic analysis (B) as well as the obtained complete proteomic data with the main statistical thresholds ​​used (C) and a general assessment of obtained protein groups highlighting their biological functions (cellular antioxidant defense/cell survival and differentiation/cellular inflammatory response/protein homeostasis) and their percentage of identified protein numbers (for each protein group) compared to number of total identified proteins (D). “Significant” is used for the proteins whose intensities are statistically significantly changed between experimental cell groups according to one-way ANOVA. “Non-significant” is used for the proteins whose intensities aren’t changed statistically significantly between cell groups according to one-way ANOVA. (hCaspase1, human caspase-1; LFQ, label-free quantification; FDR, false discovery rate; nanoHPLC, nano-high-performance liquid chromatography; MS, mass spectrometry; QexactiveOrbiTrap, Q Exactive HF mass spectrometer).
5) Line 200. It is difficult to interpret what a 0.05-fold change in protein level is describing. I suggest expressing the protein expression as a percentage of the UVA irradiation group.
Answer
It has been corrected as follows:
The C. amblystomatis lipid extract reduced the UVA-increased level of aldo-keto reductase family 1 member A1 by 95%.
6) Line 215. The change is either statistically significant, or it is not. Rather than stating "the most significant changes", state "the largest changes".
Answer
It has been corrected according to the Reviewer suggestion.:
The proteomic dataset revealed the largest changes in proteins with the cytoprotective activity against oxidative stress, as illustrated in Figure 5.
7) The Results section is very long. Inclusion of Supplementary Table 3 in the main body of the text would help greatly and reduce the amount of text needed to describe the fold changes.
Answer
Supplementary Table 3 is included in the main text as Table 2. Also, the amount of the text describing fold changes is reduced. It is shown with a strikethrough in the main text.
8) The n value is low (n=3). The data should give an indication of the variability between experiments.
Answer
Thank you for mentioning this point. It was mentioned that the low sample size (n=3) is important in evaluating the results due to the variation it brought between the samples (as limitation). However, it is also seen in the results that PCA showed an acceptable good clustering. For this reason, this situation is explained as below:
Given the small sample size (n=3), the results of the PCA should be interpreted with caution, as the observed variability between the samples may limit the robustness of the PCA in accurately capturing the full spectrum of the experimental variation. However, in our study, the 3D principal component analysis (PCA) of the proteomic data has shown good clustering of the CTR and UVA samples in distinct groups (Component 1 - 36.2%; Component 2 - 12%; Component 3 - 6.7%; Figure 2).
9) Figure 5. How do you explain the lower heme oxygenase-1 expression for C.a. compared to UVA? Is C.a. in the absence of UVA harmful?
Answer
Since the antioxidant cytoprotective cellular response is a complex and very dynamic biological process, only heme-oxygenase-1 expression lowering by the C. amblystomatis treatment in the absence of UVA exposure doesn’t mean that it will be harmful. Because, other antioxidant proteins also participate in this response. This situation should be taken into consideration and further studies analyzing cell redox status/survival are suggested. The text is improved in this direction, as given below:
In addition, without the UVA exposure, the level of heme oxygenase-1 in was lower in the C. amblystomatis lipid extract treated with fibroblasts versus control cells. That should be taken into consideration for future studies including multidimensional monitoring of cellular redox as well as cell survival status. Since the cellular antioxidant response is a complex and very dynamic biological process involving numerous proteins [28], it would not be correct to comment that the application of C. amblystomatis alone negatively affects the antioxidant response of control cells based on only dramatic changes in the heme-oxygenase 1 expression. Moreover, we have observed the level of peroxiredoxin-5 to have increased due to the application of C. amblystomatis alone.
10) Line 318. Were these pull-down assays? If not, the correlation of change in protein expression does not infer linkage of the proteins (line 324).
Answer
Thank you for indicating this important point. It is clarified in the text as below:
The heatmap displays the Pearson correlation matrix of the identified proteins with the significantly altered expression between cell groups, specifically those involved in regulating intracellular redox balance or inflammatory cellular responses (Table 2, Supplementary Figure 4) demonstrated that the proteins mentioned may correlate in expression due to indirect relationships, rather than direct interactions, due to shared regulatory factors such as transcription factors or signaling pathways.
11) What is the mechanism for altered regulation of protein expression by the lipids, noting that this is an acute exposure of cells to the lipids?
Answer
Thank you for pointing the regulatory mechanism of protein expression by the lipids. We have improved our paper to answer your question. It is shared below.
Lipid exposure rapidly alters protein expression through the complex signaling cascades (activation of kinase cascades; receptor-mediated signaling such as G-protein-coupled receptors activity changes); membrane and stress responses (alteration of membrane composition and associated changes in intracellular signaling such as transcriptional regulation; unfolded protein response according to endoplasmic reticulum stress), post-transcriptional modifications (such as lipidation, phosphorylation, ubiquitination of proteins) and targeted protein degradation) [35–39]. Together, these mechanisms enable cells to respond dynamically to lipid levels, shifting protein expression profiles to maintain intracellular homeostasis or adapt to stress condition.
12) Line 475. This reasoning is not clear. MMP-1 levels were reduced in the absence of the lipid extract.
Answer
We improved our explanation about reduced-MMP1 level by the UVA exposure. It is shared below.
This may be a dose-dependent result of UVA exposure leading a protective or adaptive response in fibroblasts by reducing MMP1 expression (protective cell adaptation at lower levels of stress – potentially – which may limit the cellular damage by reducing extracellular matrix breakdown [49]. It may be also occurred because of exposure duration of UVA. Because MMP1 exposure might be differentiate by specific time-course response (an initial reduction followed by an increase in expression later, due to dynamic interaction with tissue inhibitor of metalloproteinase, TIMP, [50]). In addition, it should be remembered that UVA can induce oxidative stress but also activate cellular antioxidant response [51]. Therefore, if the UVA exposure applied in our study triggered a relatively mild/low-level stress, this should also be taken into account. Further experiments including analyzing ROS levels, or evaluating antioxidant and transcription factor activity could help to clarify this situation.
13) Which fatty acids are present in each algal source? Can you speculate on the fatty acids that are responsible for the observed differences between the two types of lipid extract?
Answer
Both lipid extracts presenting a rich biologically active lipid composition (polyunsaturated omega-6 and omega-3 fatty acids). But there is differences between lipidomic profiles. Thus, it is possible to speculate that observed differences between the two types of lipid extract may be a result of this differences. The need for a fully quantitative content analysis for further studies is clear. The relevant section has been edited from the publication and shared below.
Although the lipid composition of the both lipid extracts show similarity which is enriched with polyunsaturated omega-6 and omega-3 fatty acids (PUFAs) showing antioxidant activity and anti-inflammatory potential, there are differences between lipidomic profiles [16,22]. It can be speculated that although differences of the effectiveness observed between the two lipid extracts may be coming from this lipidomic variation majorly, it is recommended that a full quantitative content analysis should be performed in future studies to focus on the cause of this difference.
14) Formatting inconsistencies were noted in the Reference section.
- Journals are abbreviated (e.g. ref 6) or not (e.g. ref 13).
- inconsistent page numbering (e.g. ref 11 versus ref 16).
Comments on the Quality of English Language
Grammatical errors were identified throughout the manuscript. These should be corrected.
Use of the word "Moreover" was used excessively.
Line 255. What is "closed results"?
Line 256. These are not temporal changes and so its not appropriate to infer "slowed" rate of expression.
Line 262. Replace "remarkably" with "markedly" and give the fold change.
Line 407. cell cycle progression.
Answer
All mentioned mistakes has been corrected according to the Reviewer suggestion. According to the Journal instructions, the Journal names were given in abbreviated format. And, for the Journals identifying publications by article number rather than page range, references are given by article number. Changes are marked in the text. The number of word "Moreover" was reduced by half, "closed results" was changed to "similar results", “slow down” was changed to “limit”, "remarkably" was replaced with "markedly" and fold-change information is given, and progression was added to cell cycle. Grammatical errors in the whole manuscript are corrected by a native speaker (Native English Editing Service) and the certificate is shared below.

Reviewer 2 Report
Comments and Suggestions for Authors
In the manuscript (ID: marinedrugs-3293498), authors aimed to research the effect of the microalgae Nannochloropsis oceanica and Chlorococcum amblystomatis lipid extracts on the redox-dependent metabolic changes in human skin fibroblasts induced by UVA radiation. Generally, the content of this manuscript meets the requirements of Marine Drugs. Therefore, I think this manuscript is suitable for publication in the journal of Marine Drugs after a major revision.
(1) Title:The title of this manuscript is too long. It is suggested that the author make reasonable modifications to make the title more concise.
(2) Line 2: Should be “the effects” rather “the effect”.
(3) Line 4-5: Should be “in UVA-induced human skin fibroblasts” rather “in human skin fibroblasts induced by UVA radiation”.
(4) Line 13-14: Should be “Nannochloropsis oceanica and Chlorococcum amblystomatis” rather “Nannochloropsis oceanica (N. oceanica) and Chlorococcum amblystomatis (C. amblystomatis)”.
(5) Line 14: Please provide the full name of UVA. When an abbreviation appears in the manuscript, write its full name first, and the abbreviation is written after the full name in parentheses. Subsequently, use the abbreviation consistently and do not write out the full term again.
(6) Keywords: Suggest the author to add this keywords of “Lipid extracts” and “cytoprotective effect”.
(7) Introduction, Line 34-36: In order for readers to understand more about the background of this study, the authors need to provide a more detailed explanation of UV radiation, such as according to the wavelength, ultraviolet radiation can be divided into: UVA, UVB, and UVC with the UV wavelengths of 315-400, 280-315 and 200-280 nm, respectively (https://doi.org/10.3390/md21020105).
(8) Introduction, Line 46-50: In recent years, marine-derived active substances have shown promising potential uses in the treatment of UV damage, such as antioxidant peptide ETT from Isochrysis zhanjiangensis, peptides from Skipjack tuna cardiac arterial bulbs, antioxidant peptides from Skipjack tuna (Katsuwonus pelamis) skins, collagen peptides from bigeye tuna (Thunnus obesus) skin and bone, gelatin from cartilage of Siberian sturgeon (Acipenser baerii), etc. It is suggested that the authors systematically review these marine bioactive ingredients, so as to further explain the innovation, importance and significance of this study.
(9) Introduction, Line 52-53: Should be “Nannochloropsis oceanica, and freshwater species, such as Chlorococcum amblystomatis” rather “Nannochloropsis oceanica (N. oceanica), and freshwater species like, such as Chlorococcum amblystomatis (C. amblystomatis)”.
(10) Introduction, Line 53: Should be “polyunsaturated fatty acids (PUFAs) from phospholipids” rather “polyunsaturated fatty acids, PUFAs, from phospholipids”.
(11) Introduction, Line 63: Should be “N. oceanica” rather “N.oceanica”.
(12) Introduction, Line 73-74: Should be “1,1-diphenyl-2-picrylhydrazyl (DPPH) and 2,2′-azino-bis-3-ethylbenzthiazoline-6-sulphonic acid (ABTS) radical scavenging assays” rather “1,1-diphenyl-2-picrylhydrazyl, DPPH, and 2,2′-azino-bis-3-ethylbenzthiazoline-6-sulphonic acid, ABTS, assays”.
(13) Line 233: Should be “in the literatures” rather “in the literature”.
(14) Line 241: Should be “2 μg/ml” rather “2μg/ml”. In addition, there are similar errors in other parts of the manuscript, and authors are advised to check the whole manuscript carefully and correct these minor errors.
(15) Discussion: There are a lot of contents in this part, which is not conducive to readers' reading. It is suggested that the author can make a mechanism diagram according to the content of this study, so as to facilitate readers' understanding of this study. In addition, it is suggested that the authors divide the discussion into different parts according to the research content, such as intracellular redox balance and inflammation, so as to make the discussion more organized.
(16) Line 595-596: Should be “N. oceanica and freshwater microalgae C. amblystomatis” rather “Nannochloropsis oceanica and freshwater microalgae Chlorococcum amblystomatis”.
(17) Line 637-638: Should be “3-[4,5-dimethylthiazol-2-yl]-2,5 diphenyl tetrazolium bromide (MTT) assay” rather “MTT (3-[4,5-dimethylthiazol-2-yl]-2,5 diphenyl tetrazolium bromide) assay”. In addition, authors are suggested to add the literature (https://doi.org/10.1039/D2FO00275B) to support the method.
(18)There are too many proper nouns or abbreviations (such as TNFα, LPS, iNOS, COX-2, ROS, MDA, ABTS, Nrf2, UV, DPPH, PCA, NF-κB, MAPKs, ERK, JNK, AP-1, PBR, FP, HILIC-MS, MS/MS, etc.) in the manuscript. The author is advised to add an abbreviation section at the end of the manuscript.
(19) The language needs to be improved both grammatically and scientifically. In addition, the writing of this manuscript is not standardized enough. It is recommended that the authors carefully revise the manuscript according to the requirements of the journal, including language issues and formatting issues.
Comments on the Quality of English Language
The language needs to be improved both grammatically and scientifically. In addition, the writing of this manuscript is not standardized enough. It is recommended that the authors carefully revise the manuscript according to the requirements of the journal, including language issues and formatting issues.
Author Response
Answers for Reviewer 2
Manuscript title: Comparison of the effect of the microalgae Nannochloropsis oceanica and Chlorococcum amblystomatis lipid extracts on the redox-dependent metabolic changes in human skin fibroblasts induced by UVA radiation
Manuscript: Marinedrugs-3293498
-------------------------------------------------------------------------------------------------------------------------------
In the manuscript (ID: marinedrugs-3293498), authors aimed to research the effect of the microalgae Nannochloropsis oceanica and Chlorococcum amblystomatis lipid extracts on the redox-dependent metabolic changes in human skin fibroblasts induced by UVA radiation. Generally, the content of this manuscript meets the requirements of Marine Drugs. Therefore, I think this manuscript is suitable for publication in the journal of Marine Drugs after a major revision.
Answer
The authors would like to thank Reviewer for preparing a very precise review of the study. All the reviewer's comments were taken into account during the improvement of the manuscript and we hope that it significantly improved its quality.
Changes to the manuscript were introduced in different colors dedicated to the comments of 3 Reviewers.
Changes prepared in response to the Reviewer 2 comments are marked in green.
(1) Title: The title of this manuscript is too long. It is suggested that the author make reasonable modifications to make the title more concise.
Answer
The title has been changed to: “Comparison of microalgae Nannochloropsis oceanica and Chlorococcum amblystomatis lipid extracts effects on UVA-induced changes in human skin fibroblasts proteome”
(2) Line 2: Should be “the effects” rather “the effect”.
Answer
As suggested by the reviewer, this has been corrected as follows
Comparison of microalgae Nannochloropsis oceanica and Chlorococcum amblystomatis lipid extracts effects on UVA-induced changes in human skin fibroblasts proteome.
(3) Line 4-5: Should be “in UVA-induced human skin fibroblasts” rather “in human skin fibroblasts induced by UVA radiation”.
Answer
It has been corrected according to the Reviewer suggestion. It is shown below.
Comparison of microalgae Nannochloropsis oceanica and Chlorococcum amblystomatis lipid extracts effects on UVA-induced changes in human skin fibroblasts proteome.
(4) Line 13-14: Should be “Nannochloropsis oceanica and Chlorococcum amblystomatis” rather “Nannochloropsis oceanica (N. oceanica) and Chlorococcum amblystomatis (C. amblystomatis)”.
(5) Line 14: Please provide the full name of UVA. When an abbreviation appears in the manuscript, write its full name first, and the abbreviation is written after the full name in parentheses. Subsequently, use the abbreviation consistently and do not write out the full term again.
Answer
They have been corrected according to the Reviewer suggestion. It is shared below.
Lipid extracts l from the microalgae Nannochloropsis oceanica and Chlorococcum amblystomatis have a great potential to prevent ultraviolet A (UVA)-induced metabolic disorders.
(6) Keywords: Suggest the author to add this keywords of “Lipid extracts” and “cytoprotective effect”.
Answer
All is added according to Reviewer suggestion, as presented below.
Keywords: Fibroblast; UVA radiation; Nannochloropsis oceanica; Chlorococcum amblystomatis; Lipid extracts; Oxidative stress; Inflammation; Proteomics; Cytoprotective effect
(7) Introduction, Line 34-36: In order for readers to understand more about the background of this study, the authors need to provide a more detailed explanation of UV radiation, such as according to the wavelength, ultraviolet radiation can be divided into: UVA, UVB, and UVC with the UV wavelengths of 315-400, 280-315 and 200-280 nm, respectively (https://doi.org/10.3390/md21020105).
Answer
Introduction is improved according to Reviewer suggestion. It is shared below.
Due to the wavelength characteristics, UV radiation is classified as follows: ultraviolet A (UVA), B (UVB) and C (UVC) with the UV wavelengths of 315–400, 280–315 and 200–280 nm, respectively [3]. However UVC is mostly absorbed by the ozone in the atmosphere [3].
(8) Introduction, Line 46-50: In recent years, marine-derived active substances have shown promising potential uses in the treatment of UV damage, such as antioxidant peptide ETT from Isochrysis zhanjiangensis, peptides from Skipjack tuna cardiac arterial bulbs, antioxidant peptides from Skipjack tuna (Katsuwonus pelamis) skins, collagen peptides from bigeye tuna (Thunnus obesus) skin and bone, gelatin from cartilage of Siberian sturgeon (Acipenser baerii), etc. It is suggested that the authors systematically review these marine bioactive ingredients, so as to further explain the innovation, importance and significance of this study.
Answer
Thank you for pointing out marine bioactive ingredients. Your contribution has been added to the introduction section with relevant references, as shown below.
In recent years, marine-derived active substances have shown promising potential uses in the treatment of UV damage, such as antioxidant peptide ETT from Isochrysis zhanjiangensis [11], peptides from Skipjack tuna cardiac arterial bulbs [12], antioxidant peptides from Skipjack tuna (Katsuwonus pelamis) skins [13], collagen peptides from bigeye tuna (Thunnus obesus) skin and bone [14], gelatin from cartilage of Siberian sturgeon (Acipenser baerii) [15], etc.
(9) Introduction, Line 52-53: Should be “Nannochloropsis oceanica, and freshwater species, such as Chlorococcum amblystomatis” rather “Nannochloropsis oceanica (N. oceanica), and freshwater species like, such as Chlorococcum amblystomatis (C. amblystomatis)”.
Answer
It has been corrected according to the Reviewer suggestion. It is shared below.
Lipid extracts obtained from microalgae of various origins have become increasingly popular, including the marine species such as Nannochloropsis oceanica, and freshwater species, such as Chlorococcum amblystomatis.
(10) Introduction, Line 53: Should be “polyunsaturated fatty acids (PUFAs) from phospholipids” rather “polyunsaturated fatty acids, PUFAs, from phospholipids”.
Answer
It has been corrected according to the Reviewer suggestion. It is shared below.
Those extracts contain bioactive lipids (polyunsaturated fatty acids (PUFAs) from phospholipids being the main vector of omega-3 (ω-3) fatty acids and glycolipids) and other lipid-soluble compounds that modulate cellular redox balance and inflammation [16,17].
(11) Introduction, Line 63: Should be “N. oceanica” rather “N.oceanica”.
Answer
It has been corrected according to the Reviewer suggestion. It is shared below.
Additionally, another study has demonstrated the protective effects of the N. oceanica lipid extract in the redox-dependent skin cells metabolism, including anti-inflammatory effects due to the reduction in the level of tumor necrosis factor alpha (TNFα), 8-iso prostaglandin F2, and 4-hydroxynonenal (4-HNE)-protein adducts [20].
(12) Introduction, Line 73-74: Should be “1,1-diphenyl-2-picrylhydrazyl (DPPH) and 2,2′-azino-bis-3-ethylbenzthiazoline-6-sulphonic acid (ABTS) radical scavenging assays” rather “1,1-diphenyl-2-picrylhydrazyl, DPPH, and 2,2′-azino-bis-3-ethylbenzthiazoline-6-sulphonic acid, ABTS, assays”.
Answer
It has been corrected according to the Reviewer suggestion. It is shared below.
(exhibited by 1,1-diphenyl-2-picrylhydrazyl (DPPH) and 2,2′-azino-bis-3-ethylbenzthiazoline-6-sulphonic acid (ABTS) radical scavenging assays)
(13) Line 233: Should be “in the literatures” rather “in the literature”.
Answer
It has been corrected according to the Reviewer suggestion. It is shared below.
Therefore, in parallel with the changes in the antioxidant protein profile, mentioned above, induced by the N. oceanica and C. amblystomatis lipid extracts, the antioxidant and anti-inflammatory potentials of which have been noted in the related literatures [18,21], the attention was also given to the expression levels of proteins involved in regulating the cellular inflammatory response.
(14) Line 241: Should be “2 μg/ml” rather “2μg/ml”. In addition, there are similar errors in other parts of the manuscript, and authors are advised to check the whole manuscript carefully and correct these minor errors.
Answer
Spaces between values and units have been inserted throughout the manuscript.
(15) Discussion: There are a lot of contents in this part, which is not conducive to readers' reading. It is suggested that the author can make a mechanism diagram according to the content of this study, so as to facilitate readers' understanding of this study. In addition, it is suggested that the authors divide the discussion into different parts according to the research content, such as intracellular redox balance and inflammation, so as to make the discussion more organized.
Answer
A diagram (Figure 7) was created according to the Reviewer suggestion and it is shared below.
Discussion is divided into 2 segments, according to Reviewer suggestion:
“3.1. Restoring intracellular redox balance disrupted by UVA irradiation” and
“3.2. Regulating inflammation through alterations in the intracellular redox balance”.
Figure 7. N. oceanica and C. amblystomatis lipid extracts restore the UVA-impaired control expression of antioxidant proteins and proteins involving inflammatory signaling in skin fibroblast (Ultraviolet A, UVA; reactive oxygen species, ROS; nuclear factor erythroid 2-related factor 2, Nrf2; nuclear factor kappa, NF-κB; heme oxygenase-1, HO-1; aldo-ketoreductase family member 1 A1, AKR1A1; thioredoxin, Trx; matrix metalloproteinase-1, MMP1; Parkinson disease protein 7, PARK7; signal transducer and activator of transcription1-alpha/beta, STAT1; macrophage migration inhibitory factor, MIF)
(16) Line 595-596: Should be “N. oceanica and freshwater microalgae C. amblystomatis” rather “Nannochloropsis oceanica and freshwater microalgae Chlorococcum amblystomatis”.
Answer
It has been corrected according to the Reviewer suggestion. It is shared below.
The marine microalgae N. oceanica and freshwater microalgae C. amblystomatis were supplied as spray-dried (powder) form by the company Allmicroalgae, Natural products S.A. located in Rua 25 de Abril s/n 2445-413 Pataias, Portugal.
(17) Line 637-638: Should be “3-[4,5-dimethylthiazol-2-yl]-2,5 diphenyl tetrazolium bromide (MTT) assay” rather “MTT (3-[4,5-dimethylthiazol-2-yl]-2,5 diphenyl tetrazolium bromide) assay”. In addition, authors are suggested to add the literature (https://doi.org/10.1039/D2FO00275B) to support the method.
Answer
It has been corrected according to the Reviewer suggestion. It is shared below.
The conditions of the cell exposure to UVA were selected at 13 J/cm3 according to the 70% cell viability appointed using 3-[4,5-dimethylthiazol-2-yl]-2,5 diphenyl tetrazolium bromide (MTT) assay [73].
(18)There are too many proper nouns or abbreviations (such as TNFα, LPS, iNOS, COX-2, ROS, MDA, ABTS, Nrf2, UV, DPPH, PCA, NF-κB, MAPKs, ERK, JNK, AP-1, PBR, FP, HILIC-MS, MS/MS, etc.) in the manuscript. The author is advised to add an abbreviation section at the end of the manuscript.
Answer
An abbreviation list is added at the end of the manuscript. It is shared below.
Abbreviation List: 1,1-diphenyl-2-picrylhydrazyl, DPPH; 2,2-azino-bis-3-ethylbenzthiazoline-6-sulphonic acid, ABTS; 4-hydroxynonenal, 4-HNE; Activator protein 1, AP-1; Aldo-ketoreductase family member 1 A1, AKR1A1; Carbon monoxide, CO; Catalase, CAT; c-jun N-terminal kinase, JNK; Cyclooxygenase-2, COX-2; Endoplasmic reticulum chaperone, BiP; Enzyme-Linked Immunosorbent Assay, ELISA; Extracellular signal-regulated kinase, ERK; Ferrous, Fe2+; Glutathione, GSH; Heat shock protein, HSP; Heme oxygenase-1, HO-1; High-resolution mass spectrometry, HILIC-MS; Immunoprecipitation, IP; Inducible nitric oxide synthase, iNOS; Interleukin, IL; Lipopolysaccharides, LPS; Macrophage migration inhibitory factor, MIF; Malondialdehyde, MDA; Mass spectrometry, MS; Matrix metalloproteinase-1, MMP1; Matrix metalloproteinase-3, MMP3; Matrix metalloproteinase-9, MMP9; Mitogen-activated protein kinases, MAPKs; Nano-high-performance liquid chromatography, nanoHPLC; Nuclear factor erythroid 2-related factor 2, Nrf2; Nuclear factor kappa, NF-κB; Omega-3, ω-3; p97-nuclear protein localization protein 4 homolog, Npl4; Parkinson disease protein 7, PARK7; Peroxiredoxin 1, PRDX1; Peroxiredoxin 2, PRDX2; Peroxiredoxin 3, PRDX3; Peroxiredoxin 4, PRDX4; Peroxiredoxin 5, PRDX5; Peroxiredoxin 6, PRDX6; Principal component analysis, PCA; Q Exactive HF mass spectrometer, Qexactive-OrbiTrap; Reactive oxygen species, ROS; Signal transducer and activator of transcription 3, STAT3; Signal transducer and activator of transcription1-alpha/beta, STAT1; Sodium dodecyl-sulfate polyacrylamide gel electrophoresis, SDS-PAGE; Src homology phosphatase-1, SHP-1; Superoxide dismutase 1/2, SOD1/2; Thioredoxin, Trx; Tissue inhibitor of metalloproteinase, TIMP; Toll-like receptor, TLR; Tumor necrosis factor alpha, TNFα; Ubiquitin-dependent degradation pathway, Ufd1; Ultraviolet, UV; Valosin-containing protein, VCP, the transitional endoplasmic reticulum ATPase p97.
(19) The language needs to be improved both grammatically and scientifically. In addition, the writing of this manuscript is not standardized enough. It is recommended that the authors carefully revise the manuscript according to the requirements of the journal, including language issues and formatting issues.
Answer
Grammatical errors were corrected by a native speaker (using Native English Editing Service). The certificate is shared below. The manuscript is revised according to the requirements of the journal, including language issues and formatting issues.

Reviewer 3 Report
Comments and Suggestions for Authors The article by Ekiner et al. refers to the field of research into the beneficial properties of microalgae for the purpose of their use in medicine and cosmetics. The article is well and clearly written and contains well-designed illustrations. The Introduction contains 19 references, of which 13 are from 2020-2024, that is, recent publications related to this area of ​​research are reflected.
The experimental design was designed to “shed light on the molecular mechanism underlying intracellular redox signaling and inflammation particularly through protein-protein interactions.” Namely, lipid extracts, intracellular redox balance and inflammation in UVA-irradiated skin fibroblasts, at the proteome level were obtained from two photosynthetic algae Nannochloropsis and Chlorococcum. The authors identified 494 proteins in the proteome, of which 309 proteins varied significantly across experimental cell groups. The experimental design is clearly demonstrated in schematic Figure 1.
The methods are described in detail in 6 points, and I think the results are reproducible if you follow the described methods.
The Figures are thoughtful and well designed. For better assimilation of the material, different colors are used on them. It is advisable to place pictures in the text after their first mention. Figures 2, 4, 5, 6 must be raised in the text.
In Figures 5 and 6, it is necessary to plot the statistical errors of the obtained data for each point.
In the Conclusion of the article, based on the obtained proteomic data, the authors concluded that lipid extracts from the studied algae reversed the suppression of UVA-induced antioxidant response radiation, and can be used for cosmetic purposes to protect the skin from UVA irradiation.
Ethical provisions are adequate. Suplements to the article are available.
Minor comment:
In the Abstract, there is no need to write in parentheses after the full name of the species their abbreviated generic name Nannochloropsis oceanica (N. oceanica) and Chlorococcum amblystomatis (C. amblystomatis). This means that if the full name of the species is given, then the following abbreviation below applies to it.
Author Response
Answers for Reviewer 3
Manuscript title: Comparison of the effect of the microalgae Nannochloropsis oceanica and Chlorococcum amblystomatis lipid extracts on the redox-dependent metabolic changes in human skin fibroblasts induced by UVA radiation
Manuscript: Marinedrugs-3293498
----------------------------------------------------------------------------------------------------------------
The article by Ekiner et al. refers to the field of research into the beneficial properties of microalgae for the purpose of their use in medicine and cosmetics. The article is well and clearly written and contains well-designed illustrations. The Introduction contains 19 references, of which 13 are from 2020-2024, that is, recent publications related to this area of ​​research are reflected.
The experimental design was designed to “shed light on the molecular mechanism underlying intracellular redox signaling and inflammation particularly through protein-protein interactions.” Namely, lipid extracts, intracellular redox balance and inflammation in UVA-irradiated skin fibroblasts, at the proteome level were obtained from two photosynthetic algae Nannochloropsis and Chlorococcum. The authors identified 494 proteins in the proteome, of which 309 proteins varied significantly across experimental cell groups. The experimental design is clearly demonstrated in schematic Figure 1.
The methods are described in detail in 6 points, and I think the results are reproducible if you follow the described methods.
The Figures are thoughtful and well designed. For better assimilation of the material, different colors are used on them. It is advisable to place pictures in the text after their first mention. Figures 2, 4, 5, 6 must be raised in the text.
In Figures 5 and 6, it is necessary to plot the statistical errors of the obtained data for each point.
In the Conclusion of the article, based on the obtained proteomic data, the authors concluded that lipid extracts from the studied algae reversed the suppression of UVA-induced antioxidant response radiation, and can be used for cosmetic purposes to protect the skin from UVA irradiation.
Ethical provisions are adequate. Suplements to the article are available.
Answers
The authors would like to thank Reviewer for preparing a very precise review of the study. All the reviewer's comments were taken into account during the improvement of the manuscript and we hope that it significantly improved its quality. A list of corrected items is below and all changes are highlighted in blue in the main text.
All figures are moved right after their first mention in the text. Thank you also for drawing attention to the standard errors in Figure 5 and 6. In order not to complicate the main figures Fig. 5 and 6. (also Perseus does not allow to put the standard errors directly in generated protein profile plots), normalized data exported from Perseus as txt. File. And, profile plots (for each indicated proteins) are generated with standard errors (Mean values ± SD of three independent samples and statistically significant differences for p ≤ 0.05). Generated figures are presented as Supplementary Figure 6. and 7.
Figure 5. Profile plot generated by Perseus software platform (version 2.0.11.0), using average intensities (log-transformed and normalized by median) of proteins with cytoprotective activity against oxidative stress which their levels were significantly changed between cell groups (CTR, control cells cultured in the standard growing medium; N.o., cells cultured in the standard growing medium containing also lipid extract from N. oceanica (2µg/ml) for 24h; C.a., cells cultured in the standard growing medium containing lipid extracts from C. amblystomatis (2µg/ml) for 24h; UVA, cells exposed to UVA (365 nm) at a dose of 13 J/cm3 and then incubated in the standard growing medium for 24h; UVA+N.o., cells exposed to UVA (365 nm) at a dose of 13 J/cm3 and then incubated in the standard growing medium containing lipid extracts from N. oceanica (2µg/ml) for 24h; UVA+C.a., cells exposed to UVA (365 nm) at a dose of 13 J/cm3 and then incubated in the standard growing medium containing lipid extracts from C. amblystomatis (2µg/ml) for 24h). Only the proteins showing at least 3-fold changes (log2FC) were highlighted with pink, blue, green, and yellow colors (different than grey color). Except for thioredoxin, N. oceanica or C. amblystomatis lipid extracts were able to reverse UVA-disrupted expression of aldo-keto reductase family member A1, peroxiredoxin-5, mitochondrial, and heme oxygenase 1, towards their CTR-level expressions. Statistics for the indicated proteins (Mean values ​​± SD of three independent samples and statistically significant differences for p ≤ 0.05) are presented in Supplementary Figure 6.
Supplementary figure 6. Profile plot generated using average intensities (log-transformed and normalized by median) of proteins with cytoprotective activity against oxidative stress which their levels were significantly changed between cell groups (CTR, control cells cultured in the standard growing medium; N.o., cells cultured in the standard growing medium containing also lipid extract from N. oceanica (2µg/ml) for 24h; C.a., cells cultured in the standard growing medium containing lipid extracts from C. amblystomatis (2µg/ml) for 24h; UVA, cells exposed to UVA (365 nm) at a dose of 13 J/cm3 and then incubated in the standard growing medium for 24h; UVA+N.o., cells exposed to UVA (365 nm) at a dose of 13 J/cm3 and then incubated in the standard growing medium containing lipid extracts from N. oceanica (2µg/ml) for 24h; UVA+C.a., cells exposed to UVA (365 nm) at a dose of 13 J/cm3 and then incubated in the standard growing medium containing lipid extracts from C. amblystomatis (2µg/ml) for 24h). Mean values ± SD of three independent samples and statistically significant differences for p ≤ 0.05 are presented: (X) is used for differences vs. CTR; (Y) is used for differences between the group UVA and UVA+N.o. or UVA+C.a.; (Z) is used for differences between UVA+N.o. and UVA+C.a.
Figure 6. Profile plot generated by Perseus software platform (version 2.0.11.0), using average intensities (log-transformed and normalized by median) of proteins participating in the regulation of cellular inflammatory response which their levels were significantly changed between cell groups CTR, control cells cultured in the standard growing medium; N.o., cells cultured in the standard growing medium containing also lipid extract from N. oceanica (2µg/ml) for 24h; C.a., cells cultured in the standard growing medium containing lipid extracts from C. amblystomatis (2µg/ml) for 24h; UVA, cells exposed to UVA (365 nm) at a dose of 13 J/cm3 and then incubated in the standard growing medium for 24h; UVA+N.o., cells exposed to UVA (365 nm) at a dose of 13 J/cm3 and then incubated in the standard growing medium containing lipid extracts from N. oceanica (2µg/ml) for 24h; UVA+C.a., cells exposed to UVA (365 nm) at a dose of 13 J/cm3 and then incubated in the standard growing medium containing lipid extracts from C. amblystomatis (2µg/ml) for 24h). Only the proteins showing at least 3-fold changes (log2FC) were highlighted with different colors (different than grey). Statistics for the indicated proteins (Mean values ± SD of three independent samples and statistically significant differences for p ≤ 0.05) are presented in Supplementary Figure 7.
Supplementary figure 7. Profile plot generated using average intensities (log-transformed and normalized by median) of proteins participating in the regulation of cellular inflammatory response which their levels were significantly changed between cell groups (CTR, control cells cultured in the standard growing medium; N.o., cells cultured in the standard growing medium containing also lipid extract from N. oceanica (2µg/ml) for 24h; C.a., cells cultured in the standard growing medium containing lipid extracts from C. amblystomatis (2µg/ml) for 24h; UVA, cells exposed to UVA (365 nm) at a dose of 13 J/cm3 and then incubated in the standard growing medium for 24h; UVA+N.o., cells exposed to UVA (365 nm) at a dose of 13 J/cm3 and then incubated in the standard growing medium containing lipid extracts from N. oceanica (2µg/ml) for 24h; UVA+C.a., cells exposed to UVA (365 nm) at a dose of 13 J/cm3 and then incubated in the standard growing medium containing lipid extracts from C. amblystomatis (2µg/ml) for 24h). Mean values ± SD of three independent samples and statistically significant differences for p ≤ 0.05 are presented: (X) is used for differences vs. CTR; (Y) is used for differences between the group UVA and UVA+N.o. or UVA+C.a.; (Z) is used for differences between UVA+N.o. and UVA+C.a.
Minor comment:
In the Abstract, there is no need to write in parentheses after the full name of the species their abbreviated generic name Nannochloropsis oceanica (N. oceanica) and Chlorococcum amblystomatis (C. amblystomatis). This means that if the full name of the species is given, then the following abbreviation below applies to it.
Answer
It has been corrected according to the Reviewer suggestion. Species abbreviations in parentheses have been deleted from the text, as noted below:
Abstract: Lipid extracts l from the microalgae Nannochloropsis oceanica and Chlorococcum amblystomatis have a great potential to prevent Ultraviolet A (UVA)-induced metabolic disorders.

Round 2
Reviewer 2 Report
Comments and Suggestions for Authors
The authors have answered my questions well and made the necessary changes to the manuscript (ID: marinedrugs-3293498). It looks ready for publication as far as I can tell. Then, I think that the manuscript can be accepted for publication in Marine Drugs.